# Offline Learning and Forgetting for Reasoning with Large Language Models

**Tianwei Ni**[1]* **, Allen Nie**[2]**, Sapana Chaudhary**[2]**, Yao Liu**[2]**, Huzefa Rangwala**[2]**, Rasool Fakoor**[2]
[1]*Mila - Quebec AI Institute & Université de Montréal,* [2]*Amazon Web Services*

**Reviewed on OpenReview:** *https://openreview.net/forum?id=RF6raEUATc*

## Abstract

Leveraging inference-time search in large language models has proven effective in further enhancing a trained model's capability to solve complex mathematical and reasoning problems. However, this approach significantly increases computational costs and inference time, as the model must generate and evaluate multiple candidate solutions to identify a viable reasoning path. To address this, we propose an effective approach that integrates search capabilities directly into the model by fine-tuning it on *unpaired* successful (*learning*) and failed reasoning paths (*forgetting*) derived from diverse search methods. A key challenge we identify is that naive fine-tuning can degrade the model's search capability; we show this can be mitigated with a smaller learning rate. Extensive experiments on the challenging Game-of-24 and Countdown arithmetic puzzles show that, replacing CoT-generated data with search-generated data for offline fine-tuning improves success rates by around 23% over inference-time search baselines, while reducing inference time by 180×. On top of this, our learning and forgetting objective consistently outperforms both supervised fine-tuning and preference-based methods.[1]

## 1 Introduction

The last few years has seen the rapid development of large language models (LLMs) (Vaswani et al., 2017; Achiam et al., 2023) and their applications to a diverse set of tasks. In particular, these LLMs have been tested on challenging benchmarks requiring high-level reasoning across domains such as mathematics (Glazer et al., 2024), abstract reasoning (Chollet et al., 2024), code generation (Zhuo et al., 2024), and science (Mitchener et al., 2025).

Chain-of-Thought (CoT) and *inference-time search* (search) have been proposed to enhance LLM generalization to reasoning problems at test time. Specifically, given a task description and input **x**, CoT (Wei et al., 2022) outputs a single reasoning path **ŷ** via greedy decoding, or samples multiple paths to compute average performance. In contrast, search-based methods (Lightman et al., 2023; Yao et al., 2023a; Snell et al., 2024) explicitly generate intermediate reasoning (possibly with multiple passes) before extracting **ŷ** via a reward model or self-evaluation. Despite inference-time search's effectiveness, it comes with greater computational costs (e.g., Fig. 1 and Chen et al. (2024c;b)).

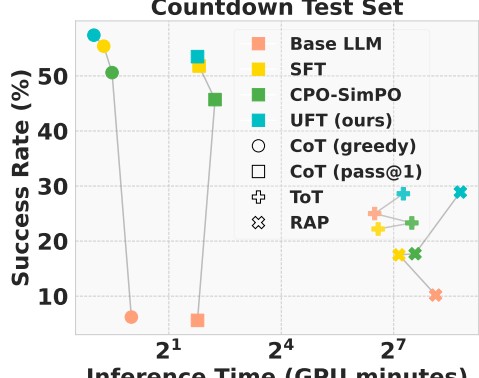

Figure 1: Trade-off between inference time and success rate in challenging arithmetic puzzles, *Countdown*. We find that fine-tuning on **CoT-style data from diverse reasoners** substantially enhances CoT inference over the base LLM (Qwen2.5-Math 7B) while preserving *inference-time search* (ToT (Yao et al., 2023a), RAP (Hao et al., 2023)) using **a smaller learning rate**. Among fine-tuning methods including SFT and preference optimization, **our method (UFT)** achieves the best CoT and search performance.

---

*Work primarily done while Tianwei Ni was an intern at Amazon. Correspondence: Tianwei Ni (`twni2016@gmail.com`), Allen Nie (`anie@cs.stanford.edu`), Rasool Fakoor (`rasool.fakoor@gmail.com`).

[1]Code is open-source at `https://github.com/twni2016/llm-reasoning-uft`.

A beneficial outcome of utilizing inference-time sampling strategies, such as CoT and search-based methods, is the generation of rich datasets. These datasets can subsequently be fed back into the model for further fine-tuning, potentially enhancing the base model's overall performance. Prior work has primarily focused on distilling the knowledge from *successful* reasoning paths, drawing from CoT-generated data (Zelikman et al., 2022; Yuan et al., 2023b; Singh et al., 2023), paths from search trees (Feng et al., 2023; Tian et al., 2024; Zhang et al., 2024a), or full search traces (Gandhi et al., 2024; Lehnert et al., 2024). However, in many complex problems, failed (reasoning) paths far outnumber successful ones, representing an underutilized resource. While potentially valuable, these failed paths are often unpaired, making them difficult to use in preference optimization (Rafailov et al., 2023). Diverse search algorithms further enrich data coverage and reasoning quality but produce heterogeneous structures (e.g., trees) unlike linear CoT traces. This raises a natural question: *can we make full use of both successful and failed reasoning paths, from both CoT and search-based methods, to improve LLM reasoning?*

In this paper, we propose a straightforward method consisting of three phases designed to enhance the reasoning capabilities of large language models: data generation, fine-tuning, and evaluation. First, we collect heterogeneous reasoning data from both CoT and the search-based methods, and convert the data into a unified CoT-style path format. Next, we fine-tune the base LLM by *learning* successful paths with the supervised fine-tuning (SFT) and *forgetting* unpaired failed paths with the unlikelihood loss. We refer to this objective as *unlikelihood fine-tuning (UFT)*. Finally, we evaluate the fine-tuned model on test sets using both CoT and inference-time search, with a primary focus on efficient CoT inference. We demonstrate the effectiveness of our three-phase pipeline on two arithmetic puzzle domains: Game-of-24 and Countdown, which are widely used as testbeds for explicit search (Yao et al., 2023a; Gandhi et al., 2024). We summarize our key findings below, using Qwen2.5-Math 1.5B and 7B (Yang et al., 2024) as base LLMs:

1. **Data quality drives performance:** On Countdown, replacing CoT data with high-quality classic search data boosts CoT performance from 33.5% to 57.1%, significantly outperforming the inference-time search baseline (25%) with a 180× reduction in inference cost.
2. **Unpaired forgetting reliably improves performance:** Incorporating failed paths using UFT improves CoT performance by 1–2% on average, with up to 7% gains in the best case. In contrast, preference-based methods like CPO-SimPO (Xu, 2024) often underperform SFT due to the paired format.
3. **Controlled learning rate helps maintain search capability:** Naive fine-tuning can quickly degrade a model's search ability. A substantially reduced learning rate helps prevent this, making it a simple but crucial step when fine-tuning with CoT-style data.

## 2 Related Work

**LLM inference-time search.** Inference-time search has shown remarkable success across reasoning (Wang et al., 2022; Stechly et al., 2024; Hao et al., 2024; Snell et al., 2024) and planning tasks (Valmeekam et al., 2023; Zheng et al., 2024; Bohnet et al., 2024). Approaches include best-of-n sampling (Nakano et al., 2021; Wang et al., 2022), multi-agent debates (Du et al., 2023), and iterative correction via predefined rules (Bai et al., 2022) or self-refinement (Shinn et al., 2023; Madaan et al., 2024). A key direction, which this paper focuses on, integrates *classic search algorithms* with LLMs, employing BFS (Yao et al., 2023a; Xie et al., 2023), DFS (Qin et al., 2023), A* (Zhuang et al., 2023; Meng et al., 2024a; Koh et al., 2024) and MCTS (Hao et al., 2023; Zhou et al., 2023; Zhao et al., 2024; Xie et al., 2024; Gao et al., 2024). While effective, inference-time search methods remain computationally expensive: self-refinement requires multiple expensive passes, multi-agent approaches demand substantial GPU memory, and search trees grow exponentially with depth (Chen et al., 2024c).

**LLM policy distillation from synthetic data.** Synthetic data, generated by classic algorithms or LLMs, is widely used for SFT in reasoning tasks to mitigate the scarcity of human-annotated datasets (Liu et al., 2024). However, despite its abundance, synthetic responses may fail to solve the task. To address this, most prior work distills reasoning knowledge to policy (Hinton et al., 2015) by selecting only correct responses. Policy distillation strategies can be categorized based on whether the source and target data originate from CoT or search algorithms. CoT-to-CoT distillation learns CoT reasoning from correct CoT paths filtered by rewards, widely used in practice (Zelikman et al., 2022; 2024; Uesato et al., 2022; Yuan et al., 2023b; Gulcehre et al., 2023; Singh et al., 2023). Search-to-search distillation either refines the proposal policy in tree search by learning

from expert behavior (Feng et al., 2023; Tian et al., 2024; Zhang et al., 2024a) and preference pairs (Chen et al., 2024a), or learns a meta-policy by imitating entire search traces (Yang et al., 2022; Gandhi et al., 2024; Lehnert et al., 2024). Notably, Zhang et al. (2024c) performs search-to-CoT distillation using preference pairs from Tree-of-Thought without a verifier. In comparison, we focus on {CoT, search}-to-CoT distillation from diverse reasoners and distill *unpaired* positive and negative data to the policy, labeled by a rule-based verifier.

**Offline fine-tuning with negative data.** Negative data has been extensively studied in *LLM safety*, where the goal is to unlearn harmful content using objectives such as Gradient Ascent and unlikelihood training (Welleck et al., 2019; Keskar et al., 2019; Yao et al., 2023b; Zhang et al., 2024b). In preference optimization, negative examples are used to increase the *relative* likelihood of preferred responses in *paired* data $(\mathbf{x}, \mathbf{y}^+, \mathbf{y}^-)$, commonly used in alignment (Rafailov et al., 2023; Yuan et al., 2023a; Zhao et al., 2023; Hong et al., 2024; Xu et al., 2024; Meng et al., 2024b) and reasoning (Pal et al., 2024; Pang et al., 2024; Zhang et al., 2024c; Setlur et al., 2024; Chen et al., 2024a). However, this paired format discards unpaired positives or negatives, reducing data efficiency. KTO (Ethayarajh et al., 2024) addresses the limitation of relative likelihood (Tuan & Wang, 2024) by learning from unpaired data, although it requires a reference model. In LLM reasoning, negative data is also used to train a reward model (Cobbe et al., 2021; Uesato et al., 2022; Lightman et al., 2023; Feng et al., 2023; Zhang et al., 2024a; Hosseini et al., 2024) for response re-ranking and search during inference. However, this does not inherently improve the base model as a policy. Instead of requiring preference pairs or reference models, we adopt unlikelihood training (Welleck et al., 2019) to directly forget failed paths as an auxiliary loss. Finally, concurrent work (Wang et al., 2025) highlights the challenges of unlearning in reasoning tasks, namely forgetting both incorrect reasoning traces and final answers while retaining overall reasoning ability.

## 3 Preliminaries

**Reasoning task as an MDP.** We formalize a reasoning task as a token-level Markov decision process (MDP) (Sutton et al., 1998) $(\mathcal{X}, \mathcal{Y}, R, T)$. The initial state $s_0 = \mathbf{x} \in \mathcal{X}$ consists of a tokenized sequence representing the task description and input. An action $y_t \in \mathcal{Y}$ is chosen based on the current state $s_t$, which is the sequence of all previous tokens: $s_t = (s_0, y_{0:t-1})$. A terminal state $s_T$ is reached upon generating a special end-of-sequence token. The ground-truth reward function $R$ is a rule-based verifier that checks the correctness of $s_T$, yielding $r = R(s_T) \in \{0, 1\}$, where 0 means failure and 1 means success. The goal is to find a policy $\pi$ that generates an *action sequence* $y_{0:T-1}$ (denoted as $\mathbf{y}$) which maximizes the reward: $\max_{y_{0:T-1}} R(s_T) = \max_{\mathbf{y}} R(\mathbf{x}, \mathbf{y})$. We refer to the terminal state $(\mathbf{x}, \mathbf{y})$ as a (reasoning) *path*.

**LLM reasoners.** We use an LLM policy $\pi_\theta$, parameterized by $\theta$, along with a search algorithm $f$ as the reasoner. We denote this LLM reasoner as $f(\pi_\theta)$ and focus on three popular LLM reasoners (Hao et al., 2024): (1) CoT (Wei et al., 2022) that uses greedy search, (2) Tree-of-Thought (ToT) (Yao et al., 2023a) that uses beam search, (3) Reasoning-via-Planning (RAP) (Hao et al., 2023) that uses MCTS (Kocsis & Szepesvári, 2006).

CoT directly outputs one (via greedy decoding) or multiple reasoning paths (under non-zero temperature) without intermediate search process[2]. On the contrary, ToT and RAP are *inference-time search* methods, first constructing a search tree of reasoning paths before selecting a final path $(\mathbf{x}, \mathbf{y})$ as the answer. Their performance is evaluated based on the success rate of the selected reasoning path. Please see Appendix C for more details.

## 4 Fine-tuning on Unpaired Correct and Failed Paths from Diverse Reasoners

In this section, we describe our approach to solving a reasoning task, as outlined in Fig. 2. Our approach involves three stages: (1) generating reasoning data from diverse reasoners, (2) offline fine-tuning the base LLM from *unpaired* correct and failed paths, and (3) evaluating it with reasoning algorithms.

---

[2]CoT's performance is often measured by the *average* success rate over generated reasoning paths, also known as the **pass@1** metric (Chen et al., 2021; Guo et al., 2025), as used in this work.

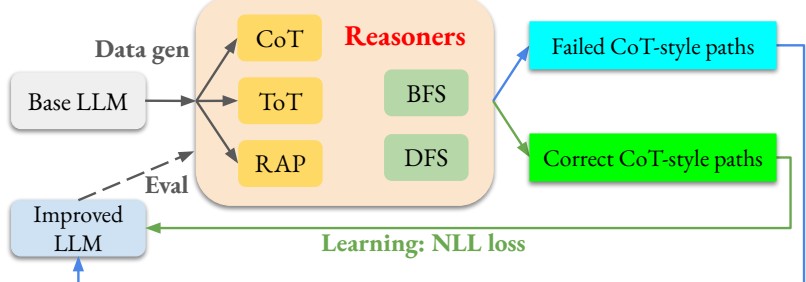

Figure 2: **Our method for reasoning tasks.** We first generate synthetic reasoning data using multiple LLM reasoners (e.g., CoT, ToT, RAP) and classic algorithms (e.g., BFS, DFS). This data is converted to a unified CoT-style format and labeled as correct or failed by a ground-truth verifier to form an unpaired dataset. We then fine-tune the base LLM with negative log-likelihood (NLL) loss on correct paths and unlikelihood loss on failed paths, which we refer to as *unlikelihood fine-tuning* (UFT). Finally, we evaluate the improved LLM with multiple LLM reasoners on a test set.

## 4.1 CoT-style Data Generation Using LLM and Classic Reasoners

For a given reasoning task, we generate synthetic reasoning data from multiple reasoners. LLM reasoners $f(\pi_\theta)$ share a common base policy $\pi_\theta$ but differ in their search algorithms $f$, which include CoT and inference-time search methods such as ToT and RAP. Optionally, reasoning paths can also be generated using classical search algorithms (*classic reasoners*) such as DFS and BFS, which rely on an external verifier rather than LLMs.

For search-based methods, we extract all root-to-leaf paths from the search tree – rather than only the final selected path – to follow the format of *CoT-style paths*. This enables (1) unifying heterogeneous search traces into a common format as training data, and (2) efficient inference, as CoT inference is fastest. We aggregate CoT-style paths from all reasoners to compose the training dataset $\mathcal{D}$:

$$\mathcal{D} = \{(\mathbf{x}, \mathbf{y}, r) \mid f \in \mathcal{F}, \mathbf{x} \in \mathcal{X}, \mathbf{y} \in f(\pi_\theta)(\mathbf{x}), r = R(\mathbf{x}, \mathbf{y})\}, \tag{1}$$

where $\mathcal{F}$ is the set of considered reasoners, and $f(\pi_\theta)(\mathbf{x})$ is the set of all reasoning paths produced by reasoner $f(\pi_\theta)$. Note that classic reasoners $f$ do not depend on $\pi_\theta$. Each path is labeled by a verifier as correct ($r = 1$) or failed ($r = 0$), splitting $\mathcal{D}$ into two unpaired datasets: the correct dataset $\mathcal{D}^+ = \{(\mathbf{x}, \mathbf{y}) \mid (\mathbf{x}, \mathbf{y}, 1) \in \mathcal{D}\}$ and the failed dataset $\mathcal{D}^- = \{(\mathbf{x}, \mathbf{y}) \mid (\mathbf{x}, \mathbf{y}, 0) \in \mathcal{D}\}$.

## 4.2 Unlikelihood Fine-Tuning on Unpaired Correct and Failed Paths

While prior work typically fine-tunes LLMs on correct CoT-generated paths (Yuan et al., 2023b; Singh et al., 2023) or on preference pairs of correct and failed responses (Pang et al., 2024; Zhang et al., 2024c), we extend these approaches in two directions: (1) incorporating diverse reasoners beyond CoT to *augment* the training data, (2) fine-tuning the model to avoid and forget failed reasoning patterns using *only* failed examples. Unlike most preference-based methods, our approach leverages *unpaired* positive or negative data, without requiring success-failure pairs for the same question.

**Learning to follow correct reasoning paths.** First, we consider the negative log-likelihood (NLL) loss, also known as supervised fine-tuning (SFT) (Ouyang et al., 2022; Cen et al., 2025), on correct reasoning paths collected from multiple reasoners:

$$\min_\theta -\mathbb{E}_{(\mathbf{x},\mathbf{y},r)\sim\mathcal{D}}[\mathbf{1}(r = 1) \log \pi_\theta(\mathbf{y} \mid \mathbf{x})] = -\mathbb{E}_{(\mathbf{x},\mathbf{y}^+)\sim\mathcal{D}^+}\left[\log \pi_\theta(\mathbf{y}^+ \mid \mathbf{x})\right] := J_{\text{NLL}}(\theta; \mathcal{D}^+). \tag{2}$$

Examining the gradient of the NLL objective (Eq. 2), $-\mathbb{E}_{(\mathbf{x},\mathbf{y},r)\sim\mathcal{D}}[\mathbf{1}(r = 1)\nabla \log \pi_\theta(\mathbf{y} \mid \mathbf{x})]$, reveals its connection to REINFORCE (Williams, 1992) with a binary (indicator) reward function, as noted in prior work (Zelikman et al., 2022; Gulcehre et al., 2023; Singh et al., 2023)[3].

---

[3]This resemblance holds when $\mathcal{D}$ is close to on-policy data (Fakoor et al., 2020), e.g., CoT-generated, but may break down as model weights shift during fine-tuning or if $\mathcal{D}$ includes search-derived data.

**Learning to avoid and forget incorrect reasoning paths.** While NLL encourages the model to learn correct reasoning paths, it can only leverage the limited number of available successful trajectories and discards a large number of failed ones – which often vastly outnumber the correct ones (e.g., $|\mathcal{D}^-|/|\mathcal{D}^+|$ ranges from 5 to 400 in our experiments). To address the limitation of NLL, we propose leveraging failed trajectories by incorporating the **unlikelihood** (UL) loss (Welleck et al., 2019) as an additional objective. The UL loss enables the model to learn to avoid (i.e., forget) failed reasoning paths, effectively utilizing the large set of failed trajectories.

$$\min_\theta J_{\text{UL}}(\theta; \mathcal{D}^-) := -\mathbb{E}_{(\mathbf{x}, \mathbf{y}^-) \sim \mathcal{D}^-}\left[\log(1 - \pi_\theta(\mathbf{y}^- \mid \mathbf{x}))\right]. \tag{3}$$

This objective explicitly reduces the probability of incorrect paths under the model. Unlike Gradient Ascent (GA) unlearning (Yao et al., 2023b)[4], UL objective has a smooth optimization landscape, with gradient:

$$\nabla J_{\text{UL}}(\theta) = \mathbb{E}_{(\mathbf{x}, \mathbf{y}^-) \sim \mathcal{D}^-}\left[\frac{\pi_\theta(\mathbf{y}^- \mid \mathbf{x})}{1 - \pi_\theta(\mathbf{y}^- \mid \mathbf{x})} \nabla \log \pi_\theta(\mathbf{y}^- \mid \mathbf{x})\right]. \tag{4}$$

The gradient implies convergence to the stationary point where $\pi_\theta(\mathbf{y}^- \mid \mathbf{x}) = 0, \forall(\mathbf{x}, \mathbf{y}^-) \in \mathcal{D}^-$. From a REINFORCE view, Eq. 4 imposes an *adaptive penalty* of $-\frac{\pi_\theta(\mathbf{y}^-|\mathbf{x})}{1-\pi_\theta(\mathbf{y}^-|\mathbf{x})}$, penalizing higher-probability failures more strongly and thus efficiently suppressing wrong paths.

**Combining learning and forgetting: unlikelihood fine-tuning (UFT).** Our final objective UFT combines NLL (Eq. 2) and UL losses (Eq. 3) with a coefficient $\alpha \in (0, 1)$:

$$\min_\theta J(\theta; \mathcal{D}, \alpha) := (1 - \alpha)J_{\text{NLL}}(\theta; \mathcal{D}^+) + \alpha J_{\text{UL}}(\theta; \mathcal{D}^-). \tag{5}$$

Since correct paths provide explicit solutions while wrong paths only rule out alternatives, we treat UL as an *auxiliary* loss by setting $\alpha$ close to 0. Moreover, as correct and failed paths may be drawn from the same search trees, they often share prefixes, introducing conflicts when $\alpha$ is large, as also noted in preference-based methods (Zhang et al., 2024c). For example, suppose a prefix sequence $(\mathbf{x}, y_{0:t})$ $(t < T-1)$ appears in both $\mathcal{D}^+$ and $\mathcal{D}^-$ with equal frequency. The gradient of Eq. 5 w.r.t. this prefix is $\left(\frac{\alpha}{1-\pi_\theta(y_{0:t}|\mathbf{x})} - 1\right) \nabla \log \pi_\theta(y_{0:t} \mid \mathbf{x})$, which has a stationary point at $\pi_\theta(y_{0:t} \mid \mathbf{x}) = 1 - \alpha$. Thus, $1 - \alpha$ sets the desired probability of the shared prefix, further motivating the need for a small $\alpha$ to avoid conflicting objectives. Unlike prior work (Zhang et al., 2024c; Setlur et al., 2024; Chen et al., 2024a) that identify high-credit steps with more computation, we sidestep this with an auxiliary loss without preference pairs.

## 4.3 Practical Algorithm

We summarize our method in Algo. 1. For a given reasoning task, we partition the initial state set $\mathcal{X}$ into training, validation and test sets, namely, $\mathcal{X}_{\text{train}}, \mathcal{X}_{\text{valid}}, \mathcal{X}_{\text{test}}$. Each subset has distinct input cases with follows the same instruction template. Our approach assumes access to a ground-truth verifier in $\mathcal{X}_{\text{train}}$ (for generating training data) and $\mathcal{X}_{\text{valid}}$ (for checkpoint selection), but not in $\mathcal{X}_{\text{test}}$ on which the final evaluation is conducted.

---

**Algorithm 1** Fine-tuning on unpaired correct and failed paths from diverse reasoners

**Require:** Reasoning task $(\mathcal{X}, \mathcal{Y}, R, T)$, base LLM $\pi_\theta$, set of reasoners $\mathcal{F}$, number of epochs $E$, batch size $B$, learning rate $\eta$, UL coefficient $\alpha$ (close to 0)

1: Generate CoT-style training data from various reasoners: $\mathcal{D} = \{(\mathbf{x}, \mathbf{y}, r) \mid f \in \mathcal{F}, \mathbf{x} \in \mathcal{X}_{\text{train}}, \mathbf{y} \in f(\pi_\theta)(\mathbf{x}), r = R(\mathbf{x}, \mathbf{y})\}$, and split $\mathcal{D}$ into correct data $\mathcal{D}^+$ and failed data $\mathcal{D}^-$.

2: Fine-tune the LLM with UFT (Eq. 5) for $E|\mathcal{D}^+|/B$ steps. In each step, sample batches $\mathcal{B}^+ \sim \mathcal{D}^+$ and $\mathcal{B}^- \sim \mathcal{D}^-$ of size $B$ each, and update:

$$\theta \leftarrow \theta - \eta[(1-\alpha)\nabla J_{\text{SFT}}(\theta; \mathcal{B}^+) + \alpha \nabla J_{\text{UL}}(\theta; \mathcal{B}^-)]. \tag{6}$$

3: Evaluate the fine-tuned LLM $\pi_\theta$ on test cases by collecting $\mathcal{D}' = \{(\mathbf{x}, \mathbf{y}, r) \mid f \in \mathcal{F}, \mathbf{x} \in \mathcal{X}_{\text{test}}, \mathbf{y} \in f(\pi_\theta)(\mathbf{x}), r = R(\mathbf{x}, \mathbf{y})\}$.

---

[4]The GA objective $\min_\theta \mathbb{E}_{(\mathbf{x}, \mathbf{y}^-) \sim \mathcal{D}^-}\left[\log \pi_\theta(\mathbf{y}^- \mid \mathbf{x})\right]$ is shown to be unstable to optimize (Zhang et al., 2024b;d), as it has an unbounded optimum $(-\infty)$ and a divergent gradient near the optimum. We provide results for GA in Sec. B.3 for completeness.

Table 1: **Performance on Countdown and Game-of-24 using Qwen2.5-Math 7B as the base model fine-tuned on Countdown.** Each cell shows (success rate / inference time) averaged with 3 seeds, where the time is measured in minutes (m) on an A100 GPU. We compare the base LLM with three fine-tuning methods (**SFT** (Ouyang et al., 2022), **CPO-SimPO** (Xu, 2024), **UFT** (ours)) under two *learning regimes* (varying learning rate and training data). SimPO (Meng et al., 2024b) is omitted due to near-zero success rate. **Bolded numbers** indicate the best success rate per row. Fig. 1 visualizes the stronger result per fine-tuning method across *learning regimes*.

| Test set & Inference | Base LLM | lr=5e-6, CoT+BFS+DFS training data | | | lr=1e-6, CoT training data | | |
|---|---|---|---|---|---|---|---|
| | | SFT ($\alpha$=0) | CPO-SimPO | UFT ($\alpha$=1e-3) | SFT ($\alpha$=0) | CPO-SimPO | UFT ($\alpha$=1e-4) |
| **Countdown** (1000 cases) | | | | | | | |
| CoT (greedy) | 6.2% / 1.0m | 55.4% / 0.6m | 50.6% / 0.7m | **57.4%** / 0.5m | 24.7% / 0.5m | 23.4% / 0.8m | 24.7% / 0.5m |
| CoT (pass@1) | 5.6% / 3.4m | 51.8% / 3.5m | 45.7% / 4.7m | **53.5%** / 3.4m | 24.1% / 2.9m | 22.1% / 5.7m | 24.7% / 3.3m |
| search: ToT | 25.0% / 90m | 2.4% / 104m | 4.1% / 61m | 0.0% / 1.9m | 22.2% / 96m | 23.3% / 179m | **28.6%** / 153m |
| search: RAP | 10.2% / 278m | 7.3% / 155m | 7.0% / 137m | 0.0% / 341m | 17.5% / 141m | 17.7% / 190m | **28.9%** / 439m |
| **Game-of-24** (100 cases) | | | | | | | |
| CoT (greedy) | 6.0% / 0.1m | 35.0% / 0.1m | 24.7% / 0.1m | **42.0%** / 0.1m | 8.0% / 0.1m | 6.7% / 0.1m | 8.3% / 0.1m |
| CoT (pass@1) | 6.0% / 0.6m | 28.3% / 0.7m | 23.0% / 1.3m | **29.7%** / 0.8m | 7.3% / 0.6m | 7.0% / 1.5m | 6.7% / 0.8m |
| search: ToT | 28.0% / 6.5m | 2.3% / 8.2m | 1.0% / 4.3m | 0.3% / 0.4m | **30.7%** / 8m | 22.7% / 13m | 24.0% / 13m |
| search: RAP | 27.0% / 8.3m | 1.3% / 13m | 3.0% / 19m | 3.7% / 29m | 27.7% / 16m | 25.7% / 42m | **33.7%** / 75m |

## 5 Experiments

We evaluate our approach on two arithmetic puzzle tasks.

**Game-of-24.** In Game-of-24[5], the goal is to use basic arithmetic operations (`+-*/`) and parentheses to combine four input numbers to obtain the number 24. Each input number can only be used once, which requires exactly three high-level reasoning steps. We follow the step-by-step response format from ToT (Yao et al., 2023a). For example, given numbers `2 9 10 12`, a correct reasoning path is `12 * 2 = 24 (left: 9 10 24)`, `10 - 9 = 1 (left: 1 24)`, `24 * 1 = 24 (left: 24)`, `Answer: (12 * 2) * (10 - 9) = 24`. Each high-level step must include the remaining numbers after the operation. We implement a process-based verifier (Uesato et al., 2022) that checks both the intermediate steps and the final answer. For evaluation, we use the same test set from Yao et al. (2023a), consisting of 100 cases. For training and validation, we use 900 distinct cases, considered easier than the test set based on the human performance.

**Countdown.** The Countdown game (Gandhi et al., 2024) extends Game-of-24 by requiring the use of four integers to obtain a specified target number beyond 24. All other rules remain the same including the step-by-step format. Following Gandhi et al. (2024), we randomly generate 500k training cases, excluding 24 as a target number. For evaluation, we randomly generate 1,000 test cases with distinct target numbers from training set. This setup allows us to test generalization to Game-of-24 when trained on Countdown data.

### 5.1 Setup

**Base LLMs.** We use Qwen2.5-Math 1.5B (Q1.5B) and Qwen2.5-Math 7B (Q7B) (Yang et al., 2024) as base LLMs. Qwen2.5-Math series were state-of-the-art open-weight *mathematical* LLMs as of December 2024.

**Step 1: Data generation setup.** We mainly build upon the LLM-reasoners repository[6] (Hao et al., 2024), which provides a unified library for all the LLM reasoners studied here. We significantly accelerate inference by batching inputs using vLLM (Kwon et al., 2023). All reported inference time in our results is based on our optimized codebase. For each LLM reasoner algorithm, we vary its hyperparameters to gather data from $\mathcal{X}_{\text{train}}$. For CoT (Wei et al., 2022), we vary the temperature between $(0.5, 0.7, 1.0)$ and top-p between $(0.7, 0.8, 0.9)$ to have 9 variants. For each variant, we use a fixed 5-shot prompt template and collect 100 paths for each case in Game-of-24 and 3 paths for Countdown. For Game-of-24, we follow LLM-reasoners to implement ToT (Yao et al., 2023a) (varying the beam size from 5 to 16) and RAP (Hao et al., 2023) (varying the exploration parameter from 1.0 to 10.0). Since Countdown is implemented in Stream-of-Search (SoS) (Gandhi et al., 2024), we use SoS code to implement classic BFS and DFS algorithms, which do not use LLMs but rely on the oracle verifier. We merge the reasoning paths generated by all the variants of the same reasoner, and then remove duplicates by exact matching, to serve as our training dataset. See Appendix C for all details.

---

[5] https://en.wikipedia.org/wiki/24_(puzzle)
[6] https://github.com/maitrix-org/llm-reasoners

**Step 2: Fine-tuning setup and baselines.** We follow the alignment handbook (Tunstall et al., 2024) to implement all fine-tuning methods on an instance with 8 A100 GPUs. All methods share the same datasets and use common hyperparameters when applicable. We fix the batch size $B$ to 128 and train $E = 10$ epochs on each dataset for Game-of-24 and $E = 2$ epochs for Countdown. We use a cosine schedule and sweep the *peak* learning rate $\eta$ over (1e-5, 5e-6, 2e-6) for Q1.5B and over (5e-6, 2e-6, 1e-6) for Q7B. We set $\alpha = 0$ to have the **SFT** baseline and sweep $\alpha$ over (1e-3, 1e-4, 1e-5, 1e-6) for our UFT. We also implement two **preference-based baselines**: **SimPO** (Meng et al., 2024b), which optimizes over *paired* preference data constructed from successful and failed reasoning paths, and **CPO-SimPO** (Xu, 2024), which augments SimPO with NLL term to better support reasoning tasks. Both methods are reference-free and leverage negative data as UFT. Notably, CPO-SimPO closely resembles the non-iterative variant of RPO (Pang et al., 2024). See Appendix A for details and Table 4 for method comparison.

**Step 3: Evaluation on CoT and search-based reasoning.** Since our training dataset are all CoT-style reasoning paths, fine-tuning on these data should directly improve CoT reasoning capability. This is measured by performing *zero-shot* CoT (via greedy decoding or sampling) on $\mathcal{X}_{\text{test}}$ with fine-tuned LLM. On the other hand, we also aim to retain search-based reasoning capability after fine-tuning. This is measured by performing ToT or RAP on $\mathcal{X}_{\text{test}}$.

**Inference-time baselines on CoT and search-based reasoning.** Inference-time baselines refer to reasoners that use base LLMs without fine-tuning. For CoT reasoning, we use the *few-shot* CoT (via greedy-decoding or sampling) on the base LLMs. For search-based reasoning, we report the *best variants* of ToT and RAP with the base LLM. The results for these baselines are shown in Table 1 for Q7B and Table 6 for Q1.5B. We find that search-based reasoning (especially ToT) significantly improves success rates over CoT reasoning for Q7B ($6.2\% \rightarrow 25\%$ in Countdown, $6\% \rightarrow 28\%$ in Game-of-24), but not for Q1.5B ($2.2\% \rightarrow 2.2\%$ in Countdown, $2\%$ to $8\%$ in Game-of-24). However, search-based reasoning substantially increases inference time compared to CoT; for example, in Countdown, ToT and RAP are $90\times$ and $278\times$ slower than CoT (greedy decoding).

## 5.2 Boosting Chain-of-Thought Reasoning

We present CoT reasoning results *at inference time* for LLMs fine-tuned on the Countdown (Table 2) and the Game-of-24 datasets (Table 3). Similar to Ye et al. (2024), we also evaluate the models trained on Countdown on the Game-of-24 test set, because conceptually Game-of-24 is a subset of Countdown. For both tables, we use the highest learning rate (1e-5 for Q1.5B and 5e-6 for Q7B) that yields best performance in CoT reasoning, leaving further discussion on the learning rate for Sec. 5.3. For the UFT objective, we fix the coefficient $\alpha$ for the same model across different datasets, selected by the average success rate on validation set $\mathcal{X}_{\text{valid}}$. Our key findings are summarized below.

**NLL loss is essential for offline fine-tuning on reasoning tasks.** The most salient observation is that SimPO, while successful in alignment tasks using only a preference loss (Meng et al., 2024b), yields near-zero success across model sizes and datasets. In contrast, CPO-SimPO (Xu, 2024), which simply adds an NLL term to SimPO, achieves substantially better results. This highlights the critical role of NLL loss, shared by SFT, CPO-SimPO, and UFT, which explicitly encourages models to follow correct reasoning paths. These findings align with prior results in Pang et al. (2024); Pal et al. (2024).

**Incorporating high-quality training data is decisive for CoT reasoning.** Second, among methods that use NLL loss, performance is primarily determined by the quality of the training data, outweighing the effect of forgetting objective. We measure **data quality** by the *best-of-n success rate*: the fraction of *unique* solved cases out of all the training cases.[7] The best performances in Table 2 and Table 3 correspond to the datasets with the highest quality (BFS+DFS data in Countdown, CoT data in Game-of-24). In addition, when the two datasets have similar quality (BFS+DFS vs CoT+BFS+DFS in Countdown, CoT vs CoT+ToT+RAP in Game-of-24), their performances are also alike. Remarkably, fine-tuning on the high-quality classic search (BFS+DFS) data from Countdown yields significantly better performance on

---

[7]Although ToT and RAP greatly outperform CoT as inference-time methods (Table 1, Table 6) in Game-of-24, they produce lower-quality training data than CoT (Table 3), because CoT's best-of-n (n=100) success rate far exceeds its average (93% versus 6% in Q7B, 82% versus 5% in Q1.5B).

Table 2: **CoT reasoning performance of LLMs fine-tuned on Countdown.** Each row shows the averaged result (3 seeds) of an LLM fine-tuned on data from specified sources. Models are evaluated on Countdown (1000 cases) and Game-of-24 (100 cases) test sets using zero-shot CoT via greedy decoding (greedy) or sampling (pass@1). BFS+DFS data are generated by oracle reasoners without LLMs. We highlight our two contributions (incorporating search-derived data and the fine-tuning method UFT) and the best cell for each (test set, CoT inference) pair. **Bold** indicates the highest mean for each row. A superscript $^*$ marks results that are statistically significant according to a Welch's t-test ($p < 0.05$).

(a) Base LLM: Qwen2.5-Math 7B (best succ rate with search: 25% on Countdown, 28% on Game-of-24)

| | SFT | SimPO | CPO-SimPO | UFT |
|---|---|---|---|---|
| **CoT data** (quality: **37.6%**) | | | | |
| Countdown succ (greedy) | $32.5\%_{\pm0.3\%}$ | $0.0\%_{\pm0.0\%}$ | $27.8\%_{\pm0.5\%}$ | $\mathbf{33.5\%}_{\pm1.0\%}$ |
| Countdown succ (pass@1) | $32.2\%_{\pm0.6\%}$ | $0.0\%_{\pm0.0\%}$ | $25.9\%_{\pm1.3\%}$ | $\mathbf{32.5\%}_{\pm1.7\%}$ |
| Game-of-24 succ (greedy) | $16.0\%_{\pm1.0\%}$ | $0.0\%_{\pm0.0\%}$ | $11.7\%_{\pm3.1\%}$ | $\mathbf{16.7\%}_{\pm2.9\%}$ |
| Game-of-24 succ (pass@1) | $12.3\%_{\pm2.3\%}$ | $0.0\%_{\pm0.0\%}$ | $10.3\%_{\pm4.0\%}$ | $\mathbf{14.7\%}_{\pm4.6\%}$ |
| **BFS+DFS data** (quality: **86.2%**) | | | | |
| Countdown succ (greedy) | $56.1\%_{\pm0.4\%}$ | $0.0\%_{\pm0.0\%}$ | $51.4\%_{\pm0.6\%}$ | $\mathbf{57.1\%}^*_{\pm0.3\%}$ |
| Countdown succ (pass@1) | $53.1\%_{\pm2.6\%}$ | $0.0\%_{\pm0.0\%}$ | $48.6\%_{\pm1.3\%}$ | $\mathbf{54.4\%}_{\pm1.1\%}$ |
| Game-of-24 succ (greedy) | $37.0\%_{\pm1.0\%}$ | $0.0\%_{\pm0.0\%}$ | $31.7\%_{\pm2.5\%}$ | $\mathbf{39.7\%}_{\pm2.5\%}$ |
| Game-of-24 succ (pass@1) | $\mathbf{28.3\%}_{\pm4.0\%}$ | $0.0\%_{\pm0.0\%}$ | $28.3\%_{\pm4.0\%}$ | $24.7\%_{\pm5.7\%}$ |
| **CoT+BFS+DFS data** (quality: **86.9%**) | | | | |
| Countdown succ (greedy) | $55.4\%_{\pm1.9\%}$ | $0.0\%_{\pm0.0\%}$ | $50.6\%_{\pm0.6\%}$ | $\mathbf{57.4\%}_{\pm0.4\%}$ |
| Countdown succ (pass@1) | $51.8\%_{\pm2.4\%}$ | $0.0\%_{\pm0.0\%}$ | $45.7\%_{\pm0.4\%}$ | $\mathbf{53.5\%}_{\pm0.5\%}$ |
| Game-of-24 succ (greedy) | $35.0\%_{\pm1.0\%}$ | $0.0\%_{\pm0.0\%}$ | $24.7\%_{\pm3.1\%}$ | $\mathbf{42.0\%}^*_{\pm1.7\%}$ |
| Game-of-24 succ (pass@1) | $28.3\%_{\pm6.7\%}$ | $0.0\%_{\pm0.0\%}$ | $23.0\%_{\pm3.0\%}$ | $\mathbf{29.7\%}_{\pm4.0\%}$ |

(b) Base LLM: Qwen2.5-Math 1.5B (best succ rate with search: 2.2% on Countdown, 8% on Game-of-24)

| | SFT | SimPO | CPO-SimPO | UFT |
|---|---|---|---|---|
| **CoT data** (quality: **20.0%**) | | | | |
| Countdown succ (greedy) | $24.2\%_{\pm0.3\%}$ | $0.6\%_{\pm0.6\%}$ | $21.6\%_{\pm0.6\%}$ | $\mathbf{26.1\%}^*_{\pm0.6\%}$ |
| Countdown succ (pass@1) | $24.4\%_{\pm0.4\%}$ | $0.6\%_{\pm0.5\%}$ | $21.3\%_{\pm0.4\%}$ | $\mathbf{26.1\%}^*_{\pm0.4\%}$ |
| Game-of-24 succ (greedy) | $13.3\%_{\pm0.6\%}$ | $0.0\%_{\pm0.0\%}$ | $8.7\%_{\pm1.5\%}$ | $\mathbf{14.3\%}_{\pm0.6\%}$ |
| Game-of-24 succ (pass@1) | $9.3\%_{\pm2.5\%}$ | $0.0\%_{\pm0.0\%}$ | $7.3\%_{\pm2.1\%}$ | $\mathbf{14.0\%}^*_{\pm2.0\%}$ |
| **BFS+DFS data** (quality: **86.2%**) | | | | |
| Countdown succ (greedy) | $50.8\%_{\pm0.7\%}$ | $0.0\%_{\pm0.0\%}$ | $50.5\%_{\pm1.4\%}$ | $\mathbf{53.7\%}^*_{\pm1.2\%}$ |
| Countdown succ (pass@1) | $47.5\%_{\pm0.8\%}$ | $0.0\%_{\pm0.0\%}$ | $47.4\%_{\pm1.6\%}$ | $\mathbf{51.2\%}^*_{\pm1.1\%}$ |
| Game-of-24 succ (greedy) | $29.7\%_{\pm3.2\%}$ | $0.0\%_{\pm0.0\%}$ | $27.3\%_{\pm3.8\%}$ | $\mathbf{32.0\%}_{\pm4.6\%}$ |
| Game-of-24 succ (pass@1) | $\mathbf{23.0\%}_{\pm7.0\%}$ | $0.0\%_{\pm0.0\%}$ | $20.3\%_{\pm2.5\%}$ | $\mathbf{23.0\%}_{\pm5.2\%}$ |
| **CoT+BFS+DFS data** (quality: **86.3%**) | | | | |
| Countdown succ (greedy) | $51.1\%_{\pm0.9\%}$ | $0.0\%_{\pm0.0\%}$ | $48.9\%_{\pm0.8\%}$ | $\mathbf{51.5\%}_{\pm0.7\%}$ |
| Countdown succ (pass@1) | $\mathbf{47.1\%}_{\pm1.1\%}$ | $0.0\%_{\pm0.0\%}$ | $45.3\%_{\pm1.2\%}$ | $46.5\%_{\pm1.3\%}$ |
| Game-of-24 succ (greedy) | $\mathbf{26.3\%}_{\pm1.5\%}$ | $0.0\%_{\pm0.0\%}$ | $24.7\%_{\pm2.9\%}$ | $26.0\%_{\pm2.6\%}$ |
| Game-of-24 succ (pass@1) | $24.0\%_{\pm2.0\%}$ | $0.0\%_{\pm0.0\%}$ | $20.7\%_{\pm2.1\%}$ | $\mathbf{25.0\%}_{\pm5.3\%}$ |

Game-of-24 than models fine-tuned *directly* on Game-of-24 CoT data (37% vs 28.2% for Q7B and 29.7% vs 14.5% for Q1.5B). This highlights the importance of incorporating high-quality training data, rather than relying solely on CoT data, as is common practice.

**Unpaired learning and forgetting is more robust than paired one.** Lastly, we analyze the effect of forgetting loss to SFT. UFT achieves the highest mean (ignoring variance) in 26 out of 36 evaluation settings, and is statistically significantly better than baselines in 7 cases. Given the limited number of seeds (3–4), these results may underestimate the true significance, and we expect the evidence to strengthen with additional runs. In terms of magnitude, UFT yields an average improvement of **1.3%** (with a maximum of **4%**) in Table 3, and an average of **1.5%** (with a maximum of **7%**) in Table 2.

In contrast, CPO-SimPO shows mixed results. While it improves over SFT with an average gain of 1.2% and maximum of 5% in Table 3, it underperforms SFT in most scenarios in Table 2, with an average **drop** of 3.5%. We believe several factors contribute to the contrasting behavior of CPO-SimPO relative to SFT (and UFT) when fine-tuned on Countdown versus Game-of-24. First, as shown in Appendix Table 5, the **paired** preference requirement in CPO-SimPO results in the exclusion of 0.1% to 5.8% of correct data on Countdown, while no such filtering occurs for Game-of-24. Since correct data determines data quality, this reduction likely contributes to CPO-SimPO's weaker performance on Countdown. Second, the exclusion

Table 3: **CoT reasoning performance of LLMs fine-tuned *and* evaluated on Game-of-24.** Each row shows the averaged result (4 seeds) of an LLM fine-tuned on data from specified sources. We highlight our two contributions (incorporating search-derived data and the fine-tuning method UFT) and **the best cell** for each (test set, CoT inference) pair. **Bold** indicates the highest mean for each row. A superscript * marks results that are statistically significant according to a Welch's t-test (p < 0.05).

(a) Base LLM: Qwen2.5-Math 7B (best succ rate with search: 28%)

| | SFT | SimPO | CPO-SimPO | UFT |
|---|---|---|---|---|
| **CoT data** (quality: **92.9%**) | | | | |
| succ (greedy) | $28.2\%_{\pm 1.7\%}$ | $2.2\%_{\pm 1.3\%}$ | $\mathbf{28.5\%}_{\pm 1.9\%}$ | $28.2\%_{\pm 3.6\%}$ |
| succ (pass@1) | $22.5\%_{\pm 3.4\%}$ | $2.2\%_{\pm 1.5\%}$ | $\mathbf{27.5\%}^{*}_{\pm 2.4\%}$ | $23.2\%_{\pm 3.4\%}$ |
| **ToT+RAP data** (quality: **69.3%**) | | | | |
| succ (greedy) | $13.2\%_{\pm 3.6\%}$ | $0.0\%_{\pm 0.0\%}$ | $\mathbf{16.5\%}_{\pm 2.9\%}$ | $15.2\%_{\pm 2.2\%}$ |
| succ (pass@1) | $13.2\%_{\pm 2.8\%}$ | $0.0\%_{\pm 0.0\%}$ | $15.0\%_{\pm 4.1\%}$ | $\mathbf{17.2\%}_{\pm 1.0\%}$ |
| **CoT+ToT+RAP data** (quality: **95.3%**) | | | | |
| succ (greedy) | $27.5\%_{\pm 4.2\%}$ | $0.2\%_{\pm 0.5\%}$ | $26.8\%_{\pm 5.7\%}$ | $\mathbf{30.2\%}_{\pm 2.1\%}$ |
| succ (pass@1) | $24.2\%_{\pm 5.2\%}$ | $0.2\%_{\pm 0.5\%}$ | $22.8\%_{\pm 1.7\%}$ | $\mathbf{26.2\%}_{\pm 2.2\%}$ |

(b) Base LLM: Qwen2.5-Math 1.5B (best succ rate with search: 8%)

| | SFT | SimPO | CPO-SimPO | UFT |
|---|---|---|---|---|
| **CoT data** (quality: **82.1%**) | | | | |
| succ (greedy) | $14.5\%_{\pm 3.8\%}$ | $0.0\%_{\pm 0.0\%}$ | $\mathbf{18.5\%}_{\pm 1.9\%}$ | $18.2\%_{\pm 3.2\%}$ |
| succ (pass@1) | $17.8\%_{\pm 3.4\%}$ | $0.0\%_{\pm 0.0\%}$ | $17.3\%_{\pm 0.5\%}$ | $\mathbf{18.5\%}_{\pm 3.5\%}$ |
| **ToT+RAP data** (quality: **44.7%**) | | | | |
| succ (greedy) | $9.5\%_{\pm 0.6\%}$ | $0.0\%_{\pm 0.0\%}$ | $\mathbf{10.3\%}_{\pm 1.9\%}$ | $8.0\%_{\pm 1.4\%}$ |
| succ (pass@1) | $\mathbf{10.0\%}_{\pm 1.4\%}$ | $0.0\%_{\pm 0.0\%}$ | $9.5\%_{\pm 1.3\%}$ | $9.5\%_{\pm 2.6\%}$ |
| **CoT+ToT+RAP data** (quality: **87.6%**) | | | | |
| succ (greedy) | $20.0\%_{\pm 2.7\%}$ | $0.0\%_{\pm 0.0\%}$ | $\mathbf{21.8\%}_{\pm 1.7\%}$ | $20.5\%_{\pm 3.7\%}$ |
| succ (pass@1) | $17.0\%_{\pm 2.2\%}$ | $0.0\%_{\pm 0.0\%}$ | $18.0\%_{\pm 2.6\%}$ | $\mathbf{18.8\%}_{\pm 1.3\%}$ |

of unpaired failed data, along with the challenge of balancing preference and NLL objectives, may have a greater impact on the more difficult Countdown task.

## 5.3 Mitigating Forgetting in Search-based Reasoning

We investigate whether fine-tuned LLMs can retain search-based reasoning abilities (*e.g.*, ToT, RAP), a question that remains underexplored in the absence of continued pretraining. Evaluating this retention is analogous to measuring *backward transfer* (Lopez-Paz & Ranzato, 2017) in continual learning, i.e., whether performance on a prior task (search-based reasoning) degrades after fine-tuning on a new one (CoT-style data). We focus on Q7B, which exhibits strong search-based reasoning performance and a clear gap over CoT baselines (Table 1). For completeness, the results of Q1.5B is shown in Appendix B.

Although learning rate decay is employed following the alignment handbook, we find that using a peak learning rate of 2e-5, as in the original Qwen2.5-Math setup (Yang et al., 2024), leads to *catastrophic forgetting* (McCloskey & Cohen, 1989), with near-zero performance in inference-time search. We hypothesize that this degradation stems from the lack of reward modeling in CoT-style supervision, which would be essential for search-based reasoning (see Sec. C.4 for details).

**A very low (peak) learning rate is crucial to retaining search capability.** Inspired by continual learning (Mirzadeh et al., 2020), we adopt a much smaller learning rate to keep parameters within the "basic capability basin" (Chen et al., 2025), preserving pretrained knowledge. Fig. 3 and Fig. 5 shows the effects of (peak) learning rate in standard SFT on Q7B. As we reduce the learning rate from 5e-6 to 1e-6 (an exceptionally low value for SFT), we observe a clear trade-off: CoT reasoning performance declines rapidly (although it remains above the baseline), while inference-time search improves significantly (although often remains below the baseline). This behavior is similar to learning rate effects seen in reference-free preference optimization (Meng et al., 2024b).

**CoT training data is more effective for preserving search capability.** An interesting observation from Fig. 3 to Fig. 5 is that in both Countdown and Game-of-24 training settings, CoT data leads to better inference-time search than search-derived data, although CoT data has *much lower* quality compared to

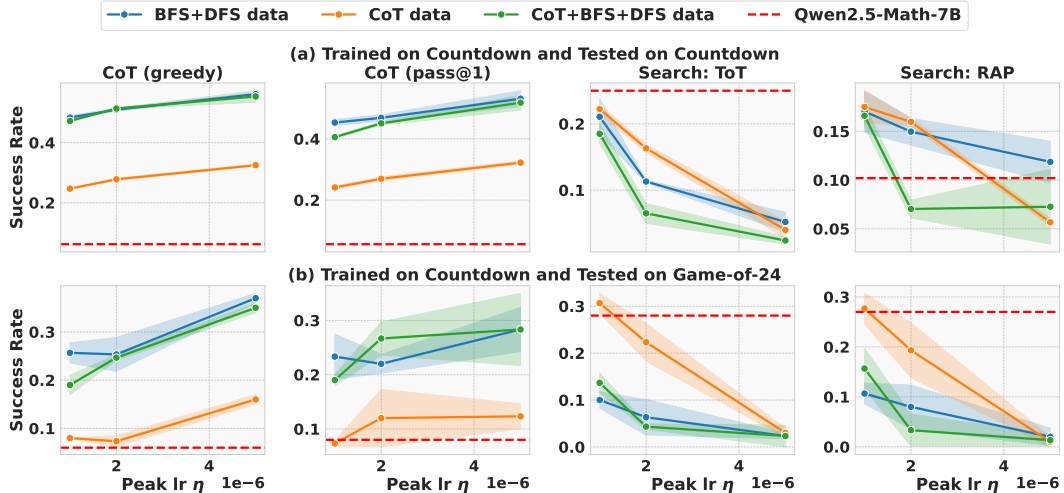

Figure 3: **Impact of (peak) learning rate and data sources on CoT vs. search-based reasoning** for standard SFT ($\alpha$=0) using Qwen2.5-Math 7B as the base model. Learning rate mediates a trade-off between CoT and search performance; CoT data better preserves search capability in most cases.

BFS+DFS data. Both CoT and search-derived data fall under CoT-style paths, but the latter is generated by a search algorithm rather than directly by the LLM. This suggests that CoT data, being more aligned with the LLM's on-policy distribution, induces less distribution shift during SFT, thereby reducing forgetting.

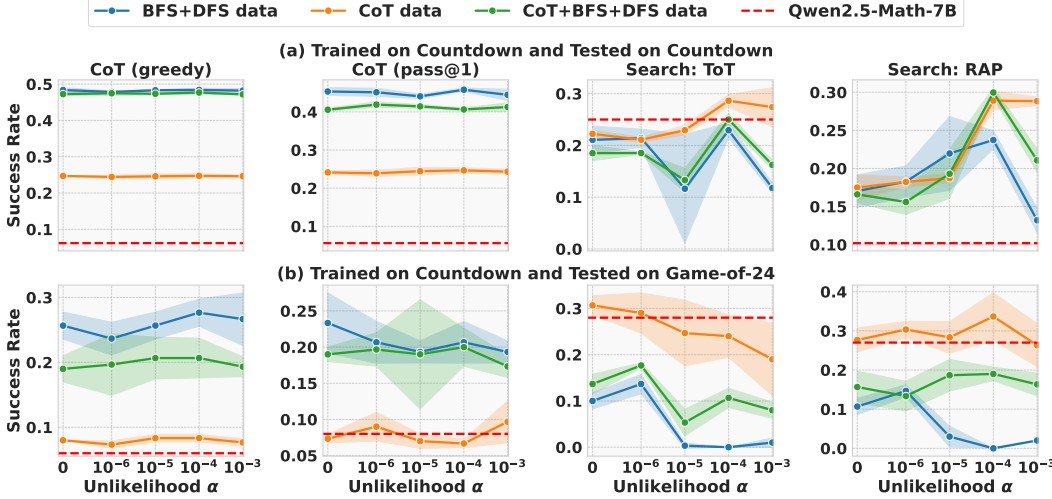

Figure 4: **Effects of unlikelihood loss and data sources on CoT vs. search-based reasoning** using Qwen2.5-Math 7B as the base model, with the peak learning rate as 1e-6. Unlikelihood loss leads to greater improvements in search performance than in CoT inference in most cases.

**Unlikelihood auxiliary loss can significantly improve search capability.** Building on the findings in Fig. 3, we observe that the effect of a low learning rate extends to UFT and CPO-SimPO as well. Here, we examine the impact of the unlikelihood loss specifically at the lowest learning rate of 1e-6. Consistent with Sec. 5.2, UFT provides marginal improvements over SFT in CoT reasoning, regardless of the choice of $\alpha$. However, with $\alpha$=1e-4, inference-time search improves considerably compared to SFT ($\alpha$=0) in most cases. For example, with the same CoT data, ToT achieves 28.6% vs. 22.2% under SFT, and RAP reaches 28.9% vs. 17.5% – far exceeding the RAP baseline of 10.2%. This indicates that learning to forget incorrect CoT-style paths could generalize to forgetting incorrect intermediate search steps.

## 6   Conclusion and Future Work

We propose a simple and effective method for improving reasoning on arithmetic puzzle domains, addressing three underexplored challenges: (1) augmenting training data by unifying mixed data formats from diverse

reasoners into a CoT-style format, (2) leveraging unpaired correct and failed paths via unlikelihood fine-tuning, and (3) mitigating loss of search capability using a small learning rate. Experiments on Countdown and Game-of-24 show that data quality, unlikelihood loss, and learning rate are key to balancing CoT efficiency and search capability. Future work can explore more advanced unlearning objectives (Tamirisa et al., 2024; Huang et al., 2024), incorporate stabilization techniques (Meng et al., 2024b), and extend the framework to other reasoning-oriented LLMs.

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

# A Fine-tuning Details

We follow the alignment handbook[8] (Tunstall et al., 2024) to fine-tune two base LLMs with full parameters, Qwen2.5-Math 1.5B (Q1.5B)[9] and Qwen2.5-Math 7B (Q7B)[10] (Yang et al., 2024). Below are common setups for all the methods (i.e., SFT, UFT, and preference-based methods).

We disable sequence packing to prevent cross-contamination[11] and set a context length of 256 tokens. which is sufficient for our tasks. Training is performed in `bfloat16` precision, and model checkpoints are saved every 5% epochs for later evaluation.

The learning rate follows `cosine_with_min_lr` schedule[12]: it linearly warms up from 0 to $\eta$ over the first 10% epochs, then decays to a minimum of 7e-8 via cosine function for the rest epochs. In Sec. 5.2, we use a peak learning rate $\eta$ of 1e-5 for Q1.5B and 5e-6 for Q7B, while in Sec. 5.3, we decrease $\eta$ to much lower values, as discussed in the main paper.

## A.1 SFT Baseline and UFT

Following Sec. 4.2, in our implementation, Unlikelihood Fine-Tuning (UFT) and Supervised Fine-Tuning (SFT) differ only by the unlikelihood loss coefficient $\alpha$. A batch size $B = 128$ is used for both successful and failed data batches (after gradient accumulation) in UFT.

## A.2 Preference-based (Reference-free) Baselines: SimPO and CPO-SimPO

**Background.** In addition to SFT, we compare UFT against offline preference-based methods: **SimPO** (Meng et al., 2024b) and **CPO-SimPO** (Xu, 2024), both designed to align LLMs with human preferences. While UFT is not a preference-based method per se, our reasoning task can be interpreted through the lens of preference alignment, where successful reasoning paths are preferred over failed ones. Like preference-based approaches, UFT learns from both successful and failed paths, making these methods natural baselines for comparison. The implementations of these methods follow the TRL library[13] (von Werra et al., 2020).

SimPO and CPO-SimPO build on DPO (Rafailov et al., 2023) and CPO (Xu et al., 2024), requiring *preference pairs* but eliminating the need for a reference model used in KL regularization (i.e., reference-free). This results in improved training efficiency and reduced GPU memory usage. By contrast, while KTO (Ethayarajh et al., 2024) like UFT operates on *unpaired* data, it still requires a reference model during training. In our preliminary experiments, running KTO using the same global batch size as UFT necessitated precomputing reference model logits and offloading the optimizer to the CPU, resulting in a 3× increase in training time on the Game-of-24 dataset and out-of-memory errors on the Countdown dataset. Consequently, we exclude KTO from our baselines and instead focus on reference-free approaches such as SimPO.

Table 4 summarizes the comparison between UFT and several offline preference-based methods. Notably, UFT is both reference-free and does not require preference pairs.

Table 4: **Brief comparison of selected offline preference-based methods and UFT.**

| Offline fine-tuning method | Preference data | Reference-free? |
|---|---|---|
| DPO (Rafailov et al., 2023), (non-iterative) RPO (Pang et al., 2024) | Paired | No |
| CPO (Xu et al., 2024), SimPO (Meng et al., 2024b), CPO-SimPO (Xu, 2024) | Paired | Yes |
| KTO (Ethayarajh et al., 2024) | Unpaired | No |
| UFT (Ours) | Unpaired | Yes |

---

[8] https://github.com/huggingface/alignment-handbook
[9] https://huggingface.co/Qwen/Qwen2.5-Math-1.5B
[10] https://huggingface.co/Qwen/Qwen2.5-Math-7B
[11] https://github.com/huggingface/transformers/issues/25452
[12] https://github.com/huggingface/transformers/blob/a22a4378d97d06b7a1d9abad6e0086d30fdea199/src/transformers/optimization.py#L338C5-L338C48
[13] https://huggingface.co/docs/trl/en/cpo_trainer

**Simple Preference Optimization with a Reference-Free Reward (SimPO) (Meng et al., 2024b).** SimPO relies on **paired** preference data, denoted as $\mathcal{D}_{\text{paired}} = \{(\mathbf{x}, \mathbf{y}^+, \mathbf{y}^-)\}$, where $(\mathbf{y}^+, \mathbf{y}^-)$ are successful and failed reasoning paths for the same input $\mathbf{x}$, respectively. To construct $\mathcal{D}_{\text{paired}}$ from our original dataset $\mathcal{D}$, we first filter for inputs that have both successful and failed paths. For each successful path $(\mathbf{x}, \mathbf{y}^+)$, we generate $E$ preference pairs by randomly sampling $E$ failed paths $(\mathbf{x}, \mathbf{y}^-)$ corresponding to the same input, treating them as rejected responses. This sampling strategy ensures that the resulting dataset is similar in size to that used for UFT and captures as much diversity as possible within the constraint of pairing.

The training objective is defined as:

$$\min_\theta J_{\text{SimPO}}(\theta; \mathcal{D}_{\text{paired}}) = -\mathbb{E}_{(\mathbf{x}, \mathbf{y}^+, \mathbf{y}^-) \sim \mathcal{D}_{\text{paired}}} \left[ \log \sigma \left( \frac{\beta}{|\mathbf{y}^+|} \log \pi_\theta(\mathbf{y}^+ \mid \mathbf{x}) - \frac{\beta}{|\mathbf{y}^-|} \log \pi_\theta(\mathbf{y}^- \mid \mathbf{x}) - \gamma \right) \right], \quad (7)$$

where $\sigma$ is the sigmoid function, $\beta = 0.1$ is a scaling factor, and $\gamma = 0.5$ is the target reward margin, following default values in TRL. The length normalization terms ($|\mathbf{y}^+|, |\mathbf{y}^-|$) help reduce bias toward longer responses. This normalization could also be explored in UFT as future work. The margin $\gamma$ enforces a minimum separation between the preferred and rejected log-probabilities.

**CPO-SimPO (Xu, 2024).** Although *pure* preference-based methods (e.g., DPO, SimPO) perform well in alignment tasks, they have been found to underperform on reasoning tasks that require explicit maximization of preferred responses (Meng et al., 2024b; Pal et al., 2024; Pang et al., 2024). While CPO-SimPO was not originally motivated by reasoning tasks, we adopt it because CPO-SimPO addresses this issue by augmenting SimPO with the NLL term from CPO (Xu et al., 2024):

$$
\begin{aligned}
&\min_\theta J_{\text{CPO-SimPO}}(\theta; \mathcal{D}_{\text{paired}}) \\
&= -\mathbb{E}_{(\mathbf{x}, \mathbf{y}^+, \mathbf{y}^-) \sim \mathcal{D}_{\text{paired}}} \left[ \log \sigma \left( \frac{\beta}{|\mathbf{y}^+|} \log \pi_\theta(\mathbf{y}^+ \mid \mathbf{x}) - \frac{\beta}{|\mathbf{y}^-|} \log \pi_\theta(\mathbf{y}^- \mid \mathbf{x}) - \gamma \right) + \lambda \log \pi_\theta(\mathbf{y}^+ \mid \mathbf{x}) \right],
\end{aligned}
\quad (8)
$$

where the NLL coefficient $\lambda = 1.0$ following the TRL.

**Connection with RPO (Pang et al., 2024).** CPO-SimPO (Xu, 2024) resembles a non-iterative version of Iterative Reasoning Preference Optimization (IRPO) (Pang et al., 2024), which we call RPO:

$$
\begin{aligned}
&\min_\theta J_{\text{RPO}}(\theta; \mathcal{D}_{\text{paired}}) \\
&= -\mathbb{E}_{(\mathbf{x}, \mathbf{y}^+, \mathbf{y}^-) \sim \mathcal{D}_{\text{paired}}} \left[ \log \sigma \left( \beta \log \frac{\pi_\theta(\mathbf{y}^+ \mid \mathbf{x})}{\pi_{\text{ref}}(\mathbf{y}^+ \mid \mathbf{x})} - \beta \log \frac{\pi_\theta(\mathbf{y}^- \mid \mathbf{x})}{\pi_{\text{ref}}(\mathbf{y}^- \mid \mathbf{x})} \right) + \frac{\lambda}{|\mathbf{y}^+|} \log \pi_\theta(\mathbf{y}^+ \mid \mathbf{x}) \right],
\end{aligned}
\quad (9)
$$

where $\pi_{\text{ref}}$ is the reference model (i.e., the base LLM). CPO-SimPO becomes equivalent to RPO when a reference model is introduced, the length normalization and reward margin in the preference term are removed, and length normalization is applied in the NLL term instead. However, since RPO, like DPO and KTO, depends on a reference model, we exclude it from our baseline comparisons.

## B  Additional Results on Countdown and Game-of-24

### B.1  Training Dataset Statistics

For Countdown, we train for 2 epochs due to the large dataset size, shown as in Table 5. On an instance with 8×A100 GPUs (40GB), total training time on CoT+BFS+DFS data is under 5 hours for Q1.5B and under 30 hours for Q7B using UFT. For Game-of-24, we train for 10 epochs. Total training time is under 0.5 hours for Q1.5B and under 2.5 hours for Q7B using UFT. SimPO and CPO-SimPO only require a single dataloader over paired data, resulting in roughly 2× shorter training time compared to UFT.

We emphasize that our goal is to reduce **inference time** in reasoning. While training may be relatively expensive, it is performed offline and does not impact deployment efficiency or user-facing latency.

Table 5: Statistics of training datasets: **total** correct paths are used by SFT and UFT, while **paired** correct paths are used by SimPO and CPO-SimPO (shown as a percentage of total). Data quality refers to the best-of-n success rate reported in Table 2 and Table 3.

| Training data (task, base, reasoner) | # Total correct paths | # Paired correct paths | Quality |
|---|---|---|---|
| Countdown, Q7B, CoT | 428.0k | 97.50% | 37.6% |
| Countdown, N/A, BFS+DFS | 614.2k | 97.55% | 86.2% |
| Countdown, Q7B, CoT+BFS+DFS | 985.3k | 99.88% | 86.9% |
| Countdown, Q1.5B, CoT | 147.9k | 94.22% | 20.0% |
| Countdown, Q1.5B, CoT+BFS+DFS | 734.3k | 99.86% | 86.3% |
| Game-of-24, Q7B, CoT | 9.1k | 100% | 92.9% |
| Game-of-24, Q7B, ToT+RAP | 5.1k | 100% | 69.3% |
| Game-of-24, Q7B, CoT+ToT+RAP | 13.7k | 100% | 95.3% |
| Game-of-24, Q1.5B, CoT | 4.4k | 100% | 82.1% |
| Game-of-24, Q1.5B, ToT+RAP | 1.7k | 100% | 44.7% |
| Game-of-24, Q1.5B, CoT+ToT+RAP | 6.0k | 100% | 87.6% |

## B.2 Additional Results with SFT and UFT

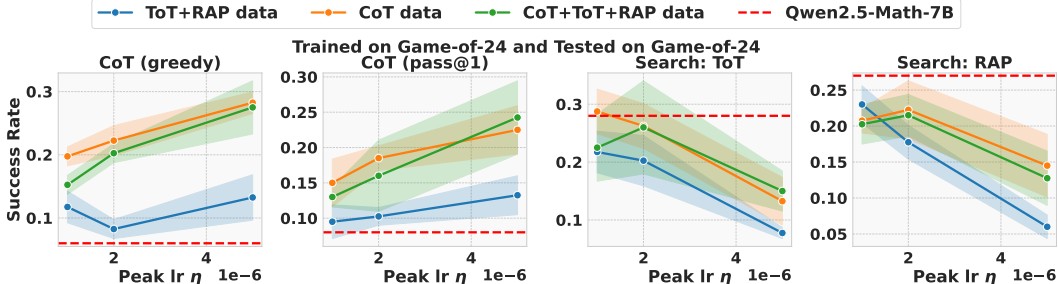

Figure 5: Results of standard SFT ($\alpha$=0) using **Qwen2.5-Math 7B** as the base model for **Game-of-24**. The same conclusion in Fig. 3 holds.

For completeness, we provide fine-tuning results based on Qwen2.5-Math 1.5B in Fig. 6 and Fig. 7. Given its weak search-based reasoning performance as a base model (e.g., success rate $\approx 2\%$ in Countdown) shown in Table 6, the effects of learning rate, CoT data, and unlikelihood loss are minimal, likely due to its poor initial capability. This suggests that the observed benefits depend on the model capability.

Table 6: Performance on Countdown and Game-of-24 test sets, using **Qwen2.5-Math 1.5B** as the base model. Due to its weak search capability, we omit fine-tuning results here (please see Table 2 and Table 3 for fine-tuning results on CoT inference).

| Test set & Inference | Base LLM |
|---|---|
| **Countdown** (1000 cases) | |
| CoT (greedy) | 2.2% / 0.6m |
| CoT (pass@1) | 2.1% / 3.2m |
| search: ToT | 2.2% / 68m |
| search: RAP | 1.4% / 143m |
| **Game-of-24** (100 cases) | |
| CoT (greedy) | 2.0% / <0.1m |
| CoT (pass@1) | 3.0% / 0.5m |
| search: ToT | 8.0% / 5.4m |
| search: RAP | 4.0% / 11.3m |

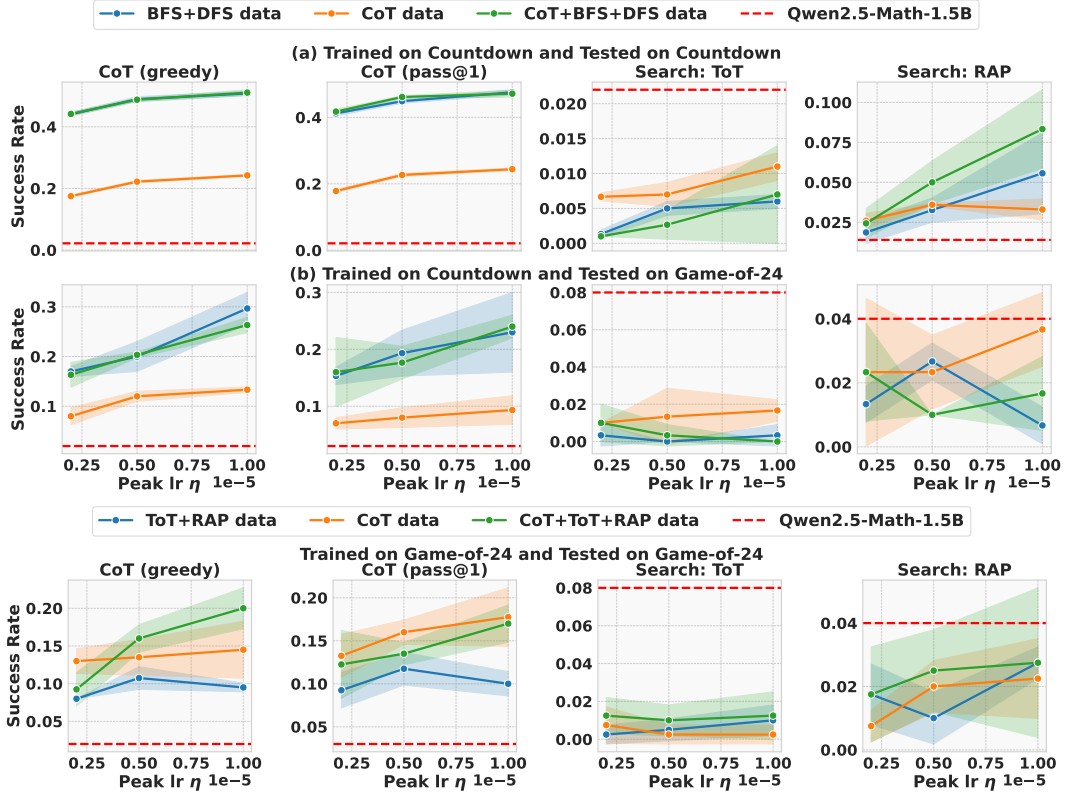

Figure 6: Results of standard SFT ($\alpha$=0) using **Qwen2.5-Math 1.5B** as the base model for **Countdown** (top) and **Game-of-24** (bottom). Due to the base model's weak search ability, learning rate and data source have little observable effect.

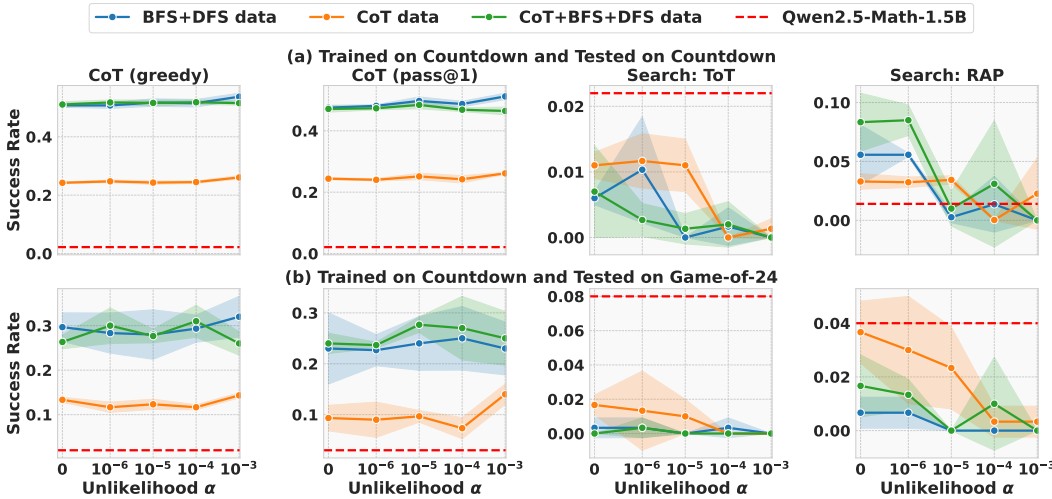

Figure 7: Results of UFT using **Qwen2.5-Math 1.5B** as the base model for **Countdown**, with the peak learning rate as 1e-5. Due to the base model's weak search ability, unlikelihood loss has little observable effect.

## B.3  Additional Results with Alternative Unlearning Methods

**Gradient Ascent and RepNoise.** To further evaluate our framework, we conduct additional experiments by replacing the unlikelihood (UL) loss $J_{\mathrm{UL}}$ in Eq. 5 with other established unlearning objectives, while keeping all other settings identical. In particular, we examine Gradient Ascent (GA) (Yao et al., 2023b) and RepNoise (Rosati et al., 2024).

GA is a simple baseline for unlearning that aims to minimize the probability of failed paths:

$$J_{\mathrm{GA}}(\theta; \mathcal{D}^-) := \mathbb{E}_{(\mathbf{x}, \mathbf{y}^-) \sim \mathcal{D}^-} \left[ \log \pi_\theta(\mathbf{y}^- \mid \mathbf{x}) \right]. \tag{10}$$

Although GA is shown to be unstable during optimization (Zhang et al., 2024b), we mitigate this issue by using a small loss coefficient of $\alpha = $ 1e-5, which is on a similar magnitude as that used for UL. This choice substantially improves numerical stability in our experiments.

Representation Noising (RepNoise) (Rosati et al., 2024), on the other hand, is a recent advanced unlearning method designed for enhancing LLM safety. It combines Gradient Ascent with removing harmful representations, encouraging the model's representations (per-layer hidden states) for negative inputs to align with random noise. Within our framework, its objective is written as:

$$
\begin{aligned}
&J_{\mathrm{RepNoise}}(\theta; \mathcal{D}^-) \\
&:= \log \left( \frac{1}{L} \mathbb{E}_{(\mathbf{x}, \mathbf{y}^-) \sim \mathcal{D}^-} \left[ \sum_{l=1}^{L} \log \pi_{\theta_{\mathrm{head}}}(\mathbf{y}^- \mid h_l^{\mathrm{post}}(\mathbf{x})) \right] \right) + \frac{\beta}{L} \mathbb{E}_{\mathbf{x} \sim \mathcal{D}^-} \left[ \sum_{l=1}^{L} \mathrm{MMD}(h_l^{\mathrm{pre}}(\mathbf{x}), \mathcal{N}(0, I)) \right].
\end{aligned}
\tag{11}
$$

The first term acts as a generalized gradient-ascent objective (with an additional log-transformation), reducing the likelihood of undesired outputs based on the per-layer post-residual hidden states $h_l^{\mathrm{post}}(\mathbf{x})$ passed into the model's output head $\pi_{\theta_{\mathrm{head}}}$, where $L = 28$ denotes the total number of layers. The second term enforces that the pre-residual hidden states $h_l^{\mathrm{pre}}(\mathbf{x})$ to be close to Gaussian noise according to the maximum mean discrepancy (MMD). Following the original implementation, we set the RepNoise loss coefficient to $\alpha = 0.5$ and its coefficient for the MMD term $\beta = $ 1e-3.

**CoT reasoning results for Gradient Ascent and RepNoise.** Table 7 shows results for CoT reasoning with the same learning rate for all unlearning methods on Game-of-24. GA achieves performance comparable to SFT and UFT, with slightly lower averages than UFT. RepNoise, which places strong emphasis on the unlearning objective (via equal weighting and representation-level forgetting), performs consistently worse than the other methods. These observations suggest that directly applying unlearning methods originally designed for LLM safety may require additional design considerations.

Across all unlearning methods, our results consistently support the claim that higher training data quality, regardless of whether the data originate from CoT or search-based algorithms, leads to better fine-tuning performance.

**Search-based reasoning results for Gradient Ascent and RepNoise.** Table 8 further examines the impact of learning rate and dataset source on CoT and search-based reasoning. Following the previous setup, we sweep learning rates over {1e-6, 2e-6, 5e-6} for the 7B base model.

For GA, our key findings for SFT and UFT (Sec. 5.3) remain valid. A smaller learning rate slightly reduces CoT reasoning performance but can substantially preserve, or even improve, the base model's search capability (e.g., 31% in ToT and 28% in RAP). Moreover, CoT training is more effective for maintaining search ability compared to mixed data (CoT+ToT+RAP), suggesting that using more on-policy data reduces distribution shift and mitigates forgetting, as discussed in Sec. 5.3.

In contrast, RepNoise exhibits a drastic drop in search-based performance (ToT and RAP), reaching 0.0% regardless of the learning rate, even though its CoT reasoning improves over the base LLM. We hypothesize that this degradation arises from the large default coefficient ($\alpha = 0.5$) in the RepNoise loss, which amplifies the effective learning rate for unlearning far beyond those used in other methods such as UFT and GA ($\leq 10^{-3}$). This aggressive unlearning, coupled with representation-level regularization toward random noise, likely disrupts internal representations crucial for search behaviors (e.g., hypothesis branching and reward estimation), even when the training objective (CoT reasoning) itself remains stable. These results underscore the importance of carefully tuning the learning rate and loss coefficient when extending unlearning objectives to LLM reasoning domains.

Table 7: **Additional results with alternative unlearning methods: CoT reasoning performance of LLMs fine-tuned and evaluated on Game-of-24.** Each cell shows the averaged result (4 seeds) of an LLM fine-tuned on data from specified sources. We highlight our two contributions (incorporating search-derived data and the fine-tuning method UFT).

(a) Base LLM: Qwen2.5-Math 7B (best succ rate with search: 28%)

| | SFT | UFT | GA | RepNoise |
|---|---|---|---|---|
| **CoT data** (quality: **92.9%**) | | | | |
| succ (greedy) | $28.2\%_{\pm1.7\%}$ | $28.2\%_{\pm3.6\%}$ | $26.8\%_{\pm2.1\%}$ | $20.5\%_{\pm3.7\%}$ |
| succ (pass@1) | $22.5\%_{\pm3.4\%}$ | $23.2\%_{\pm3.4\%}$ | $25.5\%_{\pm2.4\%}$ | $17.2\%_{\pm1.0\%}$ |
| **ToT+RAP data** (quality: **69.3%**) | | | | |
| succ (greedy) | $13.2\%_{\pm3.6\%}$ | $15.2\%_{\pm2.2\%}$ | $13.5\%_{\pm5.1\%}$ | $12.8\%_{\pm3.0\%}$ |
| succ (pass@1) | $13.2\%_{\pm2.8\%}$ | $17.2\%_{\pm1.0\%}$ | $15.8\%_{\pm2.1\%}$ | $13.2\%_{\pm5.1\%}$ |
| **CoT+ToT+RAP data** (quality: **95.3%**) | | | | |
| succ (greedy) | $27.5\%_{\pm4.2\%}$ | $30.2\%_{\pm2.1\%}$ | $28.5\%_{\pm1.7\%}$ | $20.8\%_{\pm2.1\%}$ |
| succ (pass@1) | $24.2\%_{\pm5.2\%}$ | $26.2\%_{\pm2.2\%}$ | $25.5\%_{\pm3.1\%}$ | $19.0\%_{\pm2.2\%}$ |

(b) Base LLM: Qwen2.5-Math 1.5B (best succ rate with search: 8%)

| | SFT | UFT | GA | RepNoise |
|---|---|---|---|---|
| **CoT data** (quality: **82.1%**) | | | | |
| succ (greedy) | $14.5\%_{\pm3.8\%}$ | $18.2\%_{\pm3.2\%}$ | $15.8\%_{\pm2.8\%}$ | $12.2\%_{\pm3.3\%}$ |
| succ (pass@1) | $17.8\%_{\pm3.4\%}$ | $18.5\%_{\pm3.5\%}$ | $17.0\%_{\pm2.2\%}$ | $12.0\%_{\pm2.2\%}$ |
| **ToT+RAP data** (quality: **44.7%**) | | | | |
| succ (greedy) | $9.5\%_{\pm0.6\%}$ | $8.0\%_{\pm1.4\%}$ | $9.2\%_{\pm3.8\%}$ | $5.5\%_{\pm0.6\%}$ |
| succ (pass@1) | $10.0\%_{\pm1.4\%}$ | $9.5\%_{\pm2.6\%}$ | $9.0\%_{\pm1.8\%}$ | $6.2\%_{\pm1.5\%}$ |
| **CoT+ToT+RAP data** (quality: **87.6%**) | | | | |
| succ (greedy) | $20.0\%_{\pm2.7\%}$ | $20.5\%_{\pm3.7\%}$ | $20.8\%_{\pm2.1\%}$ | $10.3\%_{\pm3.6\%}$ |
| succ (pass@1) | $17.0\%_{\pm2.2\%}$ | $18.8\%_{\pm1.3\%}$ | $20.0\%_{\pm3.4\%}$ | $12.8\%_{\pm3.6\%}$ |

Table 8: **Additional results with alternative unlearning methods: CoT and search-based reasoning performance of LLMs fine-tuned and evaluated on Game-of-24.** Each cell shows the averaged result (4 seeds) of an LLM fine-tuned on data from specified sources. We highlight the best results for each pair of training data and inference method.

(a) Gradient Ascent

| | Learning rate | CoT (greedy) | ToT | RAP |
|---|---|---|---|---|
| **Base LLM (Qwen2.5-Math 7B)** | N/A | 6.0% | 28.0% | 27.0% |
| **CoT data** (quality: **92.9%**) | 5e-6 | $\mathbf{26.8\%}_{\pm2.1\%}$ | $15.5\%_{\pm4.1\%}$ | $11.2\%_{\pm1.7\%}$ |
| | 2e-6 | $23.0\%_{\pm0.8\%}$ | $\mathbf{31.0\%}_{\pm8.8\%}$ | $22.8\%_{\pm4.5\%}$ |
| | 1e-6 | $20.2\%_{\pm1.0\%}$ | $24.2\%_{\pm3.5\%}$ | $\mathbf{28.0\%}_{\pm2.6\%}$ |
| **ToT+RAP data** (quality: **69.3%**) | 5e-6 | $\mathbf{13.5\%}_{\pm5.1\%}$ | $6.8\%_{\pm2.9\%}$ | $6.2\%_{\pm2.2\%}$ |
| | 2e-6 | $9.8\%_{\pm1.5\%}$ | $20.0\%_{\pm0.8\%}$ | $\mathbf{18.8\%}_{\pm7.6\%}$ |
| | 1e-6 | $11.8\%_{\pm1.7\%}$ | $\mathbf{20.8\%}_{\pm5.9\%}$ | $18.5\%_{\pm3.3\%}$ |
| **CoT+ToT+RAP data** (quality: **95.3%**) | 5e-6 | $\mathbf{28.5\%}_{\pm1.7\%}$ | $14.8\%_{\pm2.5\%}$ | $12.0\%_{\pm1.8\%}$ |
| | 2e-6 | $20.2\%_{\pm1.9\%}$ | $\mathbf{29.5\%}_{\pm3.3\%}$ | $\mathbf{21.8\%}_{\pm2.5\%}$ |
| | 1e-6 | $17.5\%_{\pm1.3\%}$ | $23.8\%_{\pm3.8\%}$ | $13.5\%_{\pm3.1\%}$ |

(b) RepNoise

| | Learning rate | CoT (greedy) | ToT | RAP |
|---|---|---|---|---|
| **Base LLM (Qwen2.5-Math 7B)** | N/A | 6.0% | 28.0% | 27.0% |
| **CoT data** (quality: **92.9%**) | 5e-6 | $\mathbf{20.5\%}_{\pm3.7\%}$ | $0.0\%_{\pm0.0\%}$ | $0.0\%_{\pm0.0\%}$ |
| | 2e-6 | $13.5\%_{\pm0.6\%}$ | $0.0\%_{\pm0.0\%}$ | $0.0\%_{\pm0.0\%}$ |
| | 1e-6 | $8.0\%_{\pm0.8\%}$ | $0.0\%_{\pm0.0\%}$ | $0.0\%_{\pm0.0\%}$ |
| **ToT+RAP data** (quality: **69.3%**) | 5e-6 | $\mathbf{12.8\%}_{\pm3.0\%}$ | $0.0\%_{\pm0.0\%}$ | $0.0\%_{\pm0.0\%}$ |
| | 2e-6 | $8.2\%_{\pm1.3\%}$ | $0.0\%_{\pm0.0\%}$ | $0.0\%_{\pm0.0\%}$ |
| | 1e-6 | $6.0\%_{\pm1.4\%}$ | $0.0\%_{\pm0.0\%}$ | $0.0\%_{\pm0.0\%}$ |
| **CoT+ToT+RAP data** (quality: **95.3%**) | 5e-6 | $\mathbf{20.8\%}_{\pm2.1\%}$ | $0.0\%_{\pm0.0\%}$ | $0.0\%_{\pm0.0\%}$ |
| | 2e-6 | $17.8\%_{\pm1.0\%}$ | $0.0\%_{\pm0.0\%}$ | $0.0\%_{\pm0.0\%}$ |
| | 1e-6 | $14.5\%_{\pm3.4\%}$ | $0.0\%_{\pm0.0\%}$ | $0.0\%_{\pm0.0\%}$ |

# C   Reasoner Details

## C.1   The Problems

This section provides details on the two math games not covered in the main paper.

**Game-of-24.**   We randomly split the first 900 cases (rank #1-900 by human average performance in the dataset[14]) into $\mathcal{X}_{\text{train}}$ (720 cases) and $\mathcal{X}_{\text{valid}}$ (180 cases). The $\mathcal{X}_{\text{test}}$ are the next 100 cases (rank #901-1000) following the setup in ToT (Yao et al., 2023a). As a result, the validation set evaluates in-distribution (ID) generalization, while the test set evaluates out-of-distribution (OOD) generalization based on human difficulty levels. Each case is guaranteed to be solvable using the four input numbers, which range from 1 to 13.

**Countdown.**   We follow SoS codebase[15] (Gandhi et al., 2024) to randomly generate 500k training cases, 1k validation cases, and 1k test cases. The training and validation sets share the same distribution of target numbers, while the test set includes distinct target numbers (including the number 24). Each problem consists of four input numbers sampled from 1 to 99, with a target number between 10 and 100. All generated problems are guaranteed to have at least one valid solution path using only integer intermediate results.

## C.2   CoT Reasoner

Listing 1: **Few-shot CoT template** on Countdown (5-shot prompt; only 2 shown for brevity), used in **data generation**. Here, `<input>` denotes a placeholder for the input numbers, `<target>` for the target number, and `<response>` for the LLM's output.

```
Use numbers and basic arithmetic operations (+ - * /) to obtain the target number. Each step
    , you are only allowed to choose two of the remaining numbers to obtain a new number.

Input: 25 5 5 33 Target: 27
Steps:
25 + 5 = 30 (left: 5 33 30)
30 / 5 = 6 (left: 33 6)
33 - 6 = 27 (left: 27)
Answer: 33 - ((25 + 5) / 5) = 27

Input: 45 10 11 70 Target: 94
Steps:
10 + 11 = 21 (left: 45 70 21)
45 + 70 = 115 (left: 21 115)
115 - 21 = 94 (left: 94)
Answer: (45 + 70) - (10 + 11) = 94

Input: <input> Target: <target>
Steps: <response>
```

Listing 2: **Zero-shot CoT template** on Countdown, used in **fine-tuning and evaluation**.

```
Use numbers and basic arithmetic operations (+ - * /) to obtain the target number.

Input: <input> Target: <target>
Steps: <response>
```

Chain-of-Thought (Wei et al., 2022) uses the few-shot prompting template (see Listing 1 for Countdown; Game-of-24 template follows the same format) for data generation and the zero-shot template (see Listing 2) for fine-tuning and evaluation. The "greedy-decoding" evaluation uses zero temperature. The "pass@1" evaluation uses a temperature of 0.7 and top_p of 0.8 to sample 8 paths (n=8) for Countdown and 20 paths (n=20) for Game-of-24, then calculates the average success rate.

---

[14]https://github.com/maitrix-org/llm-reasoners/blob/main/examples/ToT/game24/data/24.csv
[15]https://github.com/kanishkg/stream-of-search

### C.3 ToT and RAP Reasoners

Listing 3: **Proposed prompt template** used for search-based reasoners on Countdown (6-shot prompt; only 3 shown for brevity).

```
Perform a basic arithmetic operation (+, -, *, /) on any two of the given numbers, replacing
    them with the result. Your goal is to explore combinations that may lead to a final
    result of the target number.

Input: 25 5 5 33 Target: 27
Possible next steps:
25 + 5 = 30 (left: 5 33 30)
25 * 5 = 125 (left: 5 33 125)
25 / 5 = 5 (left: 5 33 5)
25 + 33 = 58 (left: 5 5 58)
5 * 5 = 25 (left: 25 33 25)
5 + 33 = 38 (left: 25 5 38)
33 - 25 = 8 (left: 5 5 8)
33 - 5 = 28 (left: 25 5 28)

Input: 9 25 2 Target: 43
Possible next steps:
9 + 25 = 34 (left: 34 2)
9 * 2 = 18 (left: 25 18)
9 + 2 = 11 (left: 25 11)
25 * 2 = 50 (left: 9 50)
25 + 2 = 27 (left: 9 27)

Input: 21 115 Target: 94
Possible next steps:
115 - 21 = 94 (left: 94)
115 + 21 = 136 (left: 136)

Input: <input> Target: <target>
Possible next steps: <response>
```

Listing 4: **Intermediate reward prompt template** used for search-based reasoners on Countdown (11-shot prompt; only 6 shown for brevity).

```
Evaluate if given number(s) can reach the target number (sure/likely/impossible)

Input: 27 Target: 27
sure

Input: 25 18 Target: 43
25 + 18 = 43
sure

Input: 31 10 Target: 52
31 + 10 = 41
31 - 10 = 21
31 * 10 = 310
31 / 10 = 3.1
impossible

Input: 45 70 21 Target: 94
45 + 70 + 21 = 115 + 21 = 136
-45 + 70 + 21 = 25 + 21 = 46
45 + 70 - 21 = 115 - 21 = 94
sure

Input: 15 16 16 Target: 43
15 + 16 + 16 = 47
(16 - 15) * 16 = 1 * 16 = 16
I cannot obtain 43 now, but numbers are within a reasonable range
likely

Input: 90 108 97 Target: 27
```

```
90 + 108 + 97 = 295
90 - 108 + 97 = 79
90 108 97 are all too big
impossible

Input: <input> Target: <target>
<response>
```

Listing 5: **Terminal reward prompt template** used for search-based reasoners on Countdown (6-shot prompt; only 4 shown for brevity). Here, `<answer>` is a given equation.

```
Use numbers and basic arithmetic operations (+ - * /) to obtain the target number. Given an
    input and an answer, give a judgement (sure/impossible) if the answer is correct, i.e.
    it uses each input exactly once and no other numbers, and reach the target number.

Input: 25 5 5 33 Target: 27
Answer: 33 - ((25 + 5) / 5) = 27
Judge: sure

Input: 92 91 23 54 Target: 78
Answer: 92 + 91 - (23 + 54) = 106
Judge: impossible

Input: 45 13 11 70 Target: 94
Answer: 13 + 11 + 70 = 94
Judge: impossible

Input: 25 5 5 33 Target: 27
Answer: (33 - 5) * (25 / 5) = 27
Judge: impossible

Input: <input> Target: <target>
Answer: <answer>
Judge: <response>
```

The search-based reasoners (ToT (Yao et al., 2023a) and RAP (Hao et al., 2023)) are implemented using the LLM-reasoners codebase (Hao et al., 2024). Both use the same prompt templates but differ in their underlying search algorithm.

**Tree-of-Thought (Yao et al., 2023a).** ToT employs a beam search strategy. It begins with an empty set of nodes (the beam). At each search step, the LLM is prompted using the *propose prompt* (Listing 3) to generate plausible next steps for each node in the beam. Each proposed step is then evaluated with the *intermediate reward prompt* (Listing 4), which labels responses as "sure", "likely", or "impossible". Following the original ToT setup, these labels are mapped to heuristic scores of 1, 0.1, and 0.0001, respectively. To reduce variance, we sample three responses for each reward step, and compute the average heuristic reward.

The beam is updated by selecting the top candidate nodes based on these heuristic scores, maintaining the predefined beam size. Since both Countdown and Game-of-24 tasks involve four steps in chain-of-thought reasoning, this process is repeated for four iterations. At the final search step, the *terminal reward prompt* (Listing 5) is used to assign a terminal reward to each node in the final beam. The path with the highest terminal reward is selected as the final reasoning path for evaluation, while all paths in the final beam are retained for training data.

To improve efficiency, we optimize the codebase using vLLM (Kwon et al., 2023), which enables batched inference across $\approx$1000 cases at the same time. In total, the ToT process requires $4 \times 2 = 8$ passes (i.e., 8 calls to `llm.generate()` in vLLM) per batch. This batched approach significantly accelerates the search process, achieving **over 100$\times$ speedup** compared to the original for-loop execution in the LLM-Reasoners codebase.

In practice, we vary the beam size between 5 and 16 while keeping all other search parameters fixed during training data generation. Empirically, we find that a beam size of 5 for Countdown and 6 for Game-of-24 yields the best performance on Qwen2.5-Math 7B, while beam sizes of 8 and 10 perform best for the respective

tasks on Qwen2.5-Math 1.5B. Accordingly, we adopt these optimal beam sizes for evaluating both the base LLMs and the improved models fine-tuned from them.

**Reasoning-via-Planning (Hao et al., 2023).** RAP employs a Monte Carlo Tree Search (MCTS) strategy, maintaining a search tree initialized as empty. At each iteration, a node is selected for expansion using the Upper Confidence Bound applied to Trees (UCT) algorithm (Kocsis & Szepesvári, 2006), where the exploration parameter balances exploitation and exploration. The selected node is then expanded by simulating a full rollout: the *propose prompt* generates plausible next steps, and *reward prompts* are used to assign scores to each step along the way. Steps are sampled proportionally to their rewards throughout the rollout. After completing the rollout, the terminal reward is backpropagated through the tree to update the value estimates of the visited nodes.

We run this process for 100 iterations, resulting in up to $100 \times 4 \times 2 = 800$ passes. This is a loose upper bound, as many selected nodes are leaf nodes or near-terminal, leading to shorter rollouts in practice. While the original LLM-Reasoners implementation uses only 10 iterations, we increase it to 100 to improve performance, made feasible by our use of batched inference. At the end of the search, the path with the highest terminal reward is selected for evaluation, and all explored paths are extracted for training data.

In practice, we vary the exploration parameter over (1.0, 2.0, 4.0, 6.0, 8.0, 10.0) while keeping all other search parameters fixed during training data generation. Empirically, we find that an exploration parameter of 1.0 yields the best performance for both Countdown and Game-of-24 on Qwen2.5-Math 7B, whereas a value of 2.0 performs best for both tasks on Qwen2.5-Math 1.5B. Accordingly, we adopt these optimal exploration parameters for evaluating both the base LLMs and the improved models fine-tuned from them.

### C.4 Loss of Search Capability after CoT Fine-Tuning: A Planning Perspective

An important insight emerges when comparing the prompting strategies used in CoT reasoning versus search-based methods like ToT and RAP: although they aim to solve the same task, they frame the role of the LLM in fundamentally different ways.

CoT treats the LLM as an *open-loop controller*, generating complete reasoning paths in a single pass (Listing 1) without any feedback or evaluation. In contrast, search-based methods integrates the LLM into a *closed-loop system*, where the LLM serves two interconnected roles: a policy that proposes next steps (Listing 3) and a reward model that evaluates them (Listing 4, Listing 5). This interaction between proposal and intermediate feedback is central to structured planning and search.

This difference has meaningful consequences for fine-tuning. When an LLM is fine-tuned on CoT-style paths that strip away reward signals, it may weaken its ability to perform reward modeling, which is critical in search-based reasoning. This observation offers a planning-theoretic explanation for why models fine-tuned on CoT-style paths may struggle with inference-time search.

### C.5 Classic BFS and DFS Reasoners

We also employ classic search algorithms, Breadth-First Search (BFS) and Depth-First Search (DFS), to generate training data for the Countdown task. We adopt the implementation from SoS (Gandhi et al., 2024) without modification. These symbolic reasoners are significantly faster than LLM reasoners and achieve high success rates of 65.4% (BFS) and 81.5% (DFS) on the training set (due to the use of pruning heuristics, they do not guarantee a solution in all cases). However, unlike LLM reasoners, these methods rely on external tools such as calculators and goal checkers to guide the search process.

## D   Limitation

This paper focuses on improving *reasoning language models*. Preliminary experiments on one generic language model, Mistral 7B (Jiang et al., 2023), show that while diverse reasoners still provide benefits, small learning rate and the forgetting objective offer limited improvement. We suspect these effects depend more on the base model's reasoning ability than size, similar to observations in RL that performance is bounded by base

model capability (Yue et al., 2025). Extending our method to other reasoning-oriented LLMs is a valuable direction for future work.

# E    Results on Standardized Math Benchmarks

For completeness, we also report results on standardized math benchmarks. These evaluations are not the main focus of our work, which centers on math puzzles such as Countdown and Game-of-24.

The training problems for our standardized math experiments are drawn from the OpenThoughts-114k-math dataset[16](Guha et al., 2025), which itself is extracted from NuminaMath-CoT (Li et al., 2024). This corpus integrates problems from many high-school math competitions, including the Olympiad, MATH (Hendrycks et al., 2021), AoPS forums, and AMC/AIME competitions. From this collection, we curated a subset of 38k problems with *numerical* gold answers, which we used as our training problems.

The training CoTs are constructed from three sources: (1) human-written CoTs containing gold answers in OpenThoughts, (2) self-generated CoTs whose final answers match the gold answers, and (3) self-generated CoTs whose final answers do not match the gold answers. Since OpenThoughts, like many other public datasets, does not include negative CoTs, we generate them ourselves. To verify correctness, we employ `math-verify`[17] to check the answers wrapped by `\boxed{}` in CoTs, following common practice. The instruction used for both training and testing is: `Please reason step by step, and put your final answer within \boxed{}`. SFT is trained only on CoTs (1) and (2), while the other methods are trained with CoTs (1)–(3). It is worth noting that although CoTs in category (2) contain gold answers, their reasoning steps may still be incorrect, since no process-based verifier exists for general math problems.

Due to computational resource limits, we train each method for 2 epochs with a context length of 2048, using 4 random seeds. The training time for the 7B models is around 15 hours on 8 A100 GPUs (40GB each).

We evaluate each fine-tuned LLM on the held-out test datasets using real math competitions AMC-12 (2023 and 2024)[18]. We do not include more advanced contests in our evaluation, as they typically require substantially longer context lengths (exceeding 4k tokens (Yang et al., 2025)) and large-scale online RL (Guo et al., 2025), which go beyond the scope of offline fine-tuning considered in this work.

Table 9: **CoT reasoning performance of LLMs fine-tuned on 38k high-school math problems and evaluated on AMC-12 datasets.** Each method is trained with 4 seeds. The evaluation metrics are pass@1 success rate, estimated from 32 sampled paths for each problem.

(a) Base LLM: Qwen2.5-Math 7B

|  | Base | SFT | SimPO | CPO-SimPO | UFT |
|---|---|---|---|---|---|
| AMC 2023 | 56.3% | **61.5%**$_{\pm 0.9\%}$ | 0%$_{\pm 0\%}$ | **59.3%**$_{\pm 2.2\%}$ | **61.6%**$_{\pm 0.8\%}$ |
| AMC 2024 | 29.5% | **30.1%**$_{\pm 0.7\%}$ | 0%$_{\pm 0\%}$ | **30.6%**$_{\pm 2.2\%}$ | **29.8%**$_{\pm 0.6\%}$ |

(b) Base LLM: Qwen2.5-Math 1.5B

|  | Base | SFT | SimPO | CPO-SimPO | UFT |
|---|---|---|---|---|---|
| AMC 2023 | 38.3% | **50.0%**$_{\pm 2.2\%}$ | 0%$_{\pm 0\%}$ | **50.9%**$_{\pm 3.2\%}$ | **50.2%**$_{\pm 1.2\%}$ |
| AMC 2024 | 18.6% | **28.8%**$_{\pm 0.5\%}$ | 0%$_{\pm 0\%}$ | **28.6%**$_{\pm 0.6\%}$ | **28.7%**$_{\pm 0.5\%}$ |

Table 9 shows the results. Except for SimPO, all other methods are not statistically distinguishable. Moreover, the improvements over the base LLM are minor for all 7B models. This likely reflects the noisier nature of human-written general math training data compared to the synthetic, structured data in our math puzzles. In math competitions, test problems differ substantially in style and content from training problems across years, making it harder to generalize from failed reasoning paths. By contrast, in our math puzzles, training and test problems share the same synthetic structure, which facilitates generalization. The weaker performance

---

[16]https://huggingface.co/datasets/open-r1/OpenThoughts-114k-math
[17]https://github.com/huggingface/Math-Verify
[18]https://artofproblemsolving.com/wiki/index.php/AMC_12_Problems_and_Solutions

may also stem from our limited training setup (38k problems, 2048 context length). Nevertheless, unlike SimPO, our method UFT, which also leverages negative paths, remains as stable as SFT, underscoring its robustness under limited training conditions.

