# OpenReview forum: "Offline Learning and Forgetting for Reasoning with Large Language Models"
_TMLR — Accepted by TMLR_

### Review · Reviewer_xuWZ · 2025-07-29

**Summary Of Contributions:**

The authors propose a method to improve reasoning in LLMs. The method consists of the following steps: generate CoT data with LLMs and search-based methods (DFS, BFS), unify the format, assign reward with a process-based verifier (correct or wrong). This gives unpaired failed and correct reasoning traces. Fine-tune on these with likelihood (for correct) and unlikelihood (for incorrect) training. The authors experiment with two tasks (countdown and game-of-24) and two LLMs (qwen2.5-math 1.5B and 7B). Baselines compared to are inference-team methods on the base models as well as other finetuning methods using paired correct/failed reasoning paths. They show gains of baselines in most cases, and thoroughly investigate the factors driving this, finding that high-quality data is important. Additionally, the authors find that finetuning in these ways can lead to reduced inference-time search performance, which can be mitigated both by training on "on-policy" CoT data as well as a lower learning rate.

**Audience:**

Yes

**Audience Explanation:**

The authors propose a simple method for training on unpaired correct and failed reasoning traces, which is potentially beneficial for mathematical reasoning tasks that explicitly require search (like countdown). Additionally, the authors make some interesting observations, like that CoT data leads to better retainment of search abilities after fine-tuning.

**Broader Impact Concerns:**

N.A.

**Claims And Evidence:**

No

**Claims Explanation:**

The experimental setup is sound and the authors use the right baselines and ablations, but theres a few issues:
- countdown and game-of-24 are not two separate tasks, as the authors also note themselves, one is a subset of the other. It's difficult to make claims about "mathematical reasoning" when experiments are only done on such a narrow domain.
- Gains are mostly modest or within confidence intervals, and sometimes baselines perform methods better. Can you test significance of the difference between normal SFT and UFT for the 14/18 and 16/18 times you claim your method improves? For example, I count 10 where the confidence intervals seem overlapping in table 2.

**Requested Changes:**

- (critical) Add another mathematical reasoning task that is different from countdown, or change the wording to make it less about mathematical reasoning and more about countdown / an investigation into what factors influence successful fine-tuning (data quality, learning rate, etc).
- (critical) Change wording about the method improving over baselines to only include cases where the method statistically significantly outperforms SFT
- (would strengthen the work) Recurring spelling mistakes (e.g. "while prior work typically fine-tune LLMs .."), consider correcting the text with an LLM.

---

> ### Author Response · Authors · 2025-09-20
> **Authors' Response**
>
> Thank you for your insightful comments. We have updated our PDF with changes in red. Below are our responses to your review:
>
> > (critical) Add another mathematical reasoning task that is different from countdown, or change the wording to make it less about mathematical reasoning and more about countdown / an investigation into what factors influence successful fine-tuning (data quality, learning rate, etc).
>
> We acknowledge that our main experiments focus on arithmetic puzzles. To address this concern, we took two steps:
> - **New experiments**: We conducted additional evaluations on broader math benchmarks and found that UFT performs similarly to SFT in these domains (see Appendix Section C in the updated PDF). This supports our hypothesis that UFT is most effective in search-oriented domains.
> - **Softening our claim**: To better reflect this scope, we revised the abstract, introduction, and conclusion to use “arithmetic puzzles” instead of “mathematical reasoning.” In the introduction, we further emphasize that our contribution is the full pipeline (augmenting CoT data from search traces, forgetting failed CoT paths, and preserving search ability with small learning rates), rather than the forgetting objective alone.
>
> > (critical) Change wording about the method improving over baselines to only include cases where the method statistically significantly outperforms SFT
>
> We thank the reviewer for highlighting this. In the revised paper, we now state that UFT achieves **the highest mean (ignoring variance)** in 26/36 evaluation settings, and is **statistically significantly better** in 7 cases. We also note that using only 3–4 seeds may limit statistical power, and additional runs would likely strengthen the evidence.
>
> > (would strengthen the work) Recurring spelling mistakes (e.g. "while prior work typically fine-tune LLMs .."), consider correcting the text with an LLM.
>
> We have corrected the grammatical and spelling mistakes (e.g., “fine-tune” → “fine-tunes”) and carefully proofread the revised version.

---

> > ### Comment · Reviewer_xuWZ · 2025-10-09
> >
> > The revisions make the text more faithful to the actual results in the submission, but given that the aims of the paper are now different (improving LLM search), I would propose game of 24 and countdown (both the same task) are still not enough to convincingly cover these claims. I'd suggest adding another search task like for example a synthetic graph search task (e.g. path finding) to strengthen the results. Moreover, the addition of statistical testing confirm the improvements of the method over SFT are very marginal and since they do not transfer to other domains I wonder if they transfer to other search-based tasks.

---

> > > ### Author Response · Authors · 2025-10-09
> > > **Clarification on our contributions**
> > >
> > > We thank the reviewer for the thoughtful suggestions. We have accordingly softened our claims and focused on the domain of *arithmetic puzzles*, where Countdown is a very popular benchmark, as adopted by Stream of Search (Gandhi et al., 2024) and TinyZero (Pan, 2024). Notably, the SoS paper also focuses exclusively on the Countdown domain, which already includes over 500k problems, providing a large and diverse benchmark. We therefore believe that our current task suite sufficiently captures the core challenges of this domain, while graph search lies outside our intended scope.
> > >
> > > While we acknowledge that the improvements of UFT over SFT can be modest in CoT reasoning, our main contributions lie elsewhere. Specifically:
> > > - we introduce a framework for distilling complex search algorithms such as ToT and RAP into CoT traces, enabling faster inference and improved reasoning;
> > > - we show that a carefully tuned small learning rate helps balance CoT reasoning and the preservation of search inference, an underexplored but practical direction;
> > > - we find that the forgetting objective, while mostly marginal in CoT reasoning, is more beneficial for enhancing search inference, yielding 6% and 11% gains in ToT and RAP inference respectively (see Figure 4).
> > >
> > > Together, these contributions represent a methodological advance in reasoning over arithmetic puzzles.
> > >
> > > ### Reference
> > >
> > > Kanishk Gandhi, Denise Lee, Gabriel Grand, Muxin Liu, Winson Cheng, Archit Sharma, Noah D. Goodman. Stream of Search (SoS): Learning to Search in Language. COLM 2024.
> > >
> > > TinyZero: https://github.com/Jiayi-Pan/TinyZero

---

> > > > ### Comment · Reviewer_xuWZ · 2025-10-09
> > > >
> > > > Thank you, that means the revision addresses my requested changes

---

### Review · Reviewer_xHGN · 2025-08-21

**Summary Of Contributions:**

The paper presents Unlikelihood Fine-Tuning (UFT), a procedure that extracts CoT-style paths from different reasoners (CoT, ToT, RAP, classic BFS/DFS) and fine-tunes an LLM by maximizing likelihood on successful paths while jointly applying an unlikelihood loss to unpaired failed paths to “forget” bad trajectories.
UFT outperforms other methods for CoT distillation on Game-of-24 and Countdown problems with Qwen2.5-Math base models. The work further discusses that low learning rates are needed to preserve the base models search capabilities.

### Strengths
* The problem of distilling reasoning traces into LLMs is important.
* The experiments demonstrate empirical improvements over prior work.
* The insights on the benefits of using high-quality search-generated data and low learning rates is practically useful.

### Weaknesses
* The novelty of UFT is incremental as it combines two existing ideas (NLL and UL loss) in a straight-forward manner.
* The experimental scope is confined to two closely related arithmetic reasoning tasks. The paper does not demonstrate that the findings generalize to more diverse and complex reasoning domains.

**Audience:**

Yes

**Audience Explanation:**

The problem of distilling unpaired positive and negative reasoning traces into LLMs remains an important problem. I do think the core observations are of interest.

**Claims And Evidence:**

Yes

**Claims Explanation:**

Despite the limited evaluation scope, I do think the core findings are technically correct.

**Requested Changes:**

I think the following two changes would be critical for improving the work:
* Experimental results on more diverse reasoning problems. The evaluation would be significantly more convincing if it included some tasks that are not arithmetic puzzles like Countdown and Game-of-24.
* Evaluation data for different base models. It is unclear whether the empirical observations are specific to the Qwen2.5-Math model family or if they are more broadly applicable.

---

> ### Author Response · Authors · 2025-09-20
> **Authors' Response**
>
> Thank you for your insightful comments. We have updated our PDF with changes in red. Below are our responses to your review:
>
> > Experimental results on more diverse reasoning problems. The evaluation would be significantly more convincing if it included some tasks that are not arithmetic puzzles like Countdown and Game-of-24.
>
> We acknowledge that our main experiments focus on arithmetic puzzles. To address this concern, we took two steps:
>
> - **New experiments**: We conducted additional evaluations on broader math benchmarks and found that UFT performs similarly to SFT in these domains (see Appendix Section C in the updated PDF). This supports our hypothesis that UFT is most effective in search-oriented domains.
> - **Softening our claim**: To better reflect this scope, we revised the abstract, introduction, and conclusion to use “arithmetic puzzles” instead of “mathematical reasoning.” In the introduction, we further emphasize that our contribution is the full pipeline (augmenting CoT data from search traces, forgetting failed CoT paths, and preserving search ability with small learning rates), rather than the forgetting objective alone.
>
> > Evaluation data for different base models. It is unclear whether the empirical observations are specific to the Qwen2.5-Math model family or if they are more broadly applicable.
>
> We conducted preliminary experiments with Mistral-7B as the base model and found that, while data from diverse reasoners still provide noticeable benefits, the forgetting loss and small learning rate offered limited improvement (see Limitation, Sec. E). We believe this is related to the base model’s inherent reasoning capability, as found in recent work in RL methods [Yue et al., 2025]. Due to computational constraints, we leave extending the evaluation to other reasoning-oriented base LLMs for future work.
>
> Reference: Yang Yue, Zhiqi Chen, Rui Lu, Andrew Zhao, Zhaokai Wang, Shiji Song, and Gao Huang. Does reinforcement learning really incentivize reasoning capacity in llms beyond the base model? arXiv preprint arXiv:2504.13837, 2025

---

### Review · Reviewer_n7cM · 2025-09-11

**Summary Of Contributions:**

This paper propose a better method to utlize the unsuccessful reasoning traces.  The idea is to extend the unlearning methods from LLM safety research to unlearn the unsuccessful reasoning traces.

**Additional Comments:**

I personally don't know whether there are existing works that already explore the idea of unlearning failed reasoning traces. But the idea looks standard and straightforward and there might be other existing (or concurrent) work.

I suggest the authors to actively search and  honestly disclose whether there are existing or concurrent work explore the same idea of  unlearning failed reasoning traces.   If this is not done sufficient, I think accepting this paper will be unfair for other authors who explore the same idea and be detrimental to the research community.  I will look at other reviewers' comments and the authors response to see whether this concern is addressed.

**Audience:**

Yes

**Audience Explanation:**

Yes, at least I am very interested  and excited in this study.

**Broader Impact Concerns:**

No concerns.

**Claims And Evidence:**

Yes

**Claims Explanation:**

1. Extensive experiements are done.

2. However, I require the authors to conduct the following experiment to validate the advantage of  "unlikelihood (UL) loss" compared to several alternatives:

 **Ascend loss**. This is the most standard unlearning loss, which is basically the negative NLL (or cross entropy) loss. Basically optimizing this loss is taking one step of standard gradient ascend over the unlearning dataset.

**Reprensetation noise loss**[1]. The representation loss (See Eq. (5) in [1]) basically make the representation of the unlearning dataset to be similar the pure random Gaussian noise. In your case, you basically make the representation of  failed reasoning traces to be random.

**Tamper resistant loss[2]**. The tamper resistent loss basically make the unlearning loss to be high  and **hard to recover** even though taking extra recover steps.    See the first term in Eq. (1) of [2]

**Smooth loss[3]** Similar to tamper resistent losss, smooth loss make the unlearning loss hard to learn. In order to achieve that, smooth loss (See the second term in Eq.(1) of [3]) make the learning curve of the failed reasoning traces to be smooth.

[1] Representation Noising: A Defence Mechanism Against Harmful Finetuning

[2] Tamper-Resistant Safeguards for Open-Weight LLMs

[3] Booster: Tackling Harmful Fine-tuning for Large Language Models via Attenuating Harmful Perturbation


**All these alternatives** should be tried and performed by the authors. Also, the authors should also combines the above methods with inference time reasoning methods (ToT and RAP) to evaluate.  Otherwise, I can't agree that the contribution of proposing to use this  "unlikelihood (UL) loss" is beneficial against other unlearning loss.  Feel free to leave me a comment if you have difficulty in reproducing these four unleanring losses. I am very familar with all these unlearning losses I mention.

**Requested Changes:**

1. Conduct experiments to validate the advantage of unlikelihood (UL) loss against other unlearning loss.

I am okay with acceptance if unlikelihood indeed outperform the other four losses. Please also consider to give me the anonymous code for your implementation such that I can verify by myself.

---

> ### Author Response · Authors · 2025-09-20
> **Authors' Response**
>
> Thank you for your insightful comments and detailed suggestions. We have updated our PDF with changes in red.
>
> > Alternative unlearning losses (Ascend loss, Representation noise loss, Tamper resistant loss, Smooth loss) should be tried and performed by the authors. Also, the authors should also combine the above methods with inference time reasoning methods (ToT and RAP) to evaluate. Otherwise, I can't agree that the contribution of proposing to use this "unlikelihood (UL) loss" is beneficial against other unlearning loss.
>
> > Feel free to leave me a comment if you have difficulty in reproducing these four unlearning losses. I am very familiar with all these unlearning losses I mention. Please also consider giving me the anonymous code for your implementation such that I can verify by myself.
>
> Thank you for pointing us to these unlearning losses that could help this work.
>
> **Gradient Ascent (GA).** Given the time constraint, we tried Ascend loss during the rebuttal. While GA is a popular unlearning loss, prior work [Zhang et al., 2024] notes its instability because both the loss and gradient can be unbounded. Despite this, in a preliminary Game-of-24 experiment with the 1.5B model, GA produced results comparable to UFT when a small loss coefficient was used:
> - GA achieved 16.2% ± 2.2% (greedy decoding) and 17.5% ± 3.0% (pass@1)
> - UFT achieved 18.2% ± 3.2% (greedy decoding) and 18.5% ± 3.5% (pass@1).
>
> These findings suggest GA can be effective with tuning, though UFT may have higher averages.
>
> **Other unlearning losses.** For representation noise, tamper-resistant, and smooth losses, we agree they are promising but more complex to implement. Since you mentioned your familiarity with these methods, we would be grateful for any guidance or code pointers, which could even help us run additional experiments within the rebuttal timeframe.
>
> **Reproducibility.** To support reproducibility, we have uploaded **our anonymous code to the supplementary material**. The code includes both unlikelihood loss and GA loss implementations. An excerpt is shown below (minor details omitted):
>
> ```python
> from trl import SFTTrainer
>
> class UFTTrainer(SFTTrainer):
>     def compute_ul_loss(self, model, raw_inputs, return_outputs=False):
>         pos_loss = self._standard_cross_entropy_loss(model, raw_inputs["positive"])
>
>         if self.args.ul_loss_type == "gradient_ascent":
>             neg_loss = -1.0 * self._standard_cross_entropy_loss(
>                 model, raw_inputs["negative"]
>             )
>         elif self.args.ul_loss_type == "unlikelihood":
>             neg_loss = self._unlikelihood_loss(model, raw_inputs["negative"])
>         else:
>             raise NotImplementedError(f"unknown loss {self.args.ul_loss_type}")
>
>         loss = (1.0 - self.args.ul_alpha) * pos_loss + self.args.ul_alpha * neg_loss
>         return loss
>
>     def _unlikelihood_loss(self, model, inputs):
>         outputs = model(**inputs)
>         logits = outputs["logits"].float()
>         labels = inputs["labels"]
>         shift_logits = logits[..., :-1, :].contiguous()
>         shift_labels = labels[..., 1:].contiguous()
>         shift_logits = shift_logits.view(-1, shift_logits.shape[-1])
>         shift_labels = shift_labels.view(-1).to(shift_logits.device)
>
>         valid_mask = shift_labels != -100
>         valid_logits = shift_logits[valid_mask]
>         valid_labels = shift_labels[valid_mask]
>
>         probs = F.softmax(valid_logits, dim=-1)
>         gold_probs = probs[torch.arange(valid_labels.shape[0]), valid_labels]
>         unlikelihood_loss = -torch.log(1 - gold_probs + 1e-8)
>         return unlikelihood_loss.mean()
> ```
> where `raw_inputs["positive"]` and `raw_inputs["negative"]` are independently sampled batches from positive and negative datasets, respectively.
>
> **Clarification of contributions.** We appreciate the reviewer’s focus on the forgetting component. To put our work in context, our contributions are broader and complementary:
> - Distilling search traces into CoT data by unifying outputs from different reasoners for data augmentation.
> - Demonstrating the utility of forgetting failed paths, where UL loss serves as a simple instantiation.
> - Highlighting the importance of small learning rates to preserve search ability during CoT fine-tuning.
>
> Reference: Ruiqi Zhang, Licong Lin, Yu Bai, and Song Mei. Negative preference optimization: From catastrophic collapse to effective unlearning. COLM 2024.

---

> > ### Author Response · Authors · 2025-09-20
> > **Cont.**
> >
> > > I suggest the authors actively search and honestly disclose whether there are existing or concurrent work explore the same idea of unlearning failed reasoning traces.
> >
> > Yes, there is existing and concurrent work related to unlearning failed reasoning traces. As discussed in our related work section, several approaches frame this as preference optimization (e.g., DPO), where successful paths are preferred and failed paths are dispreferred (thus unlearned). We included SimPO as a representative baseline and observed that preference-based objectives without the SFT (learning) objective can substantially degrade fine-tuning performance.
> >
> > In addition, concurrent work by Wang et al. (2025), now cited in our paper, explicitly investigates unlearning in reasoning. They highlight the challenge of forgetting both incorrect reasoning traces and final answers while retaining reasoning ability, using representation misdirection unlearning (RMU) as their objective. However, their emphasis is on improving LLM safety rather than enhancing reasoning capability, which complements but differs from our focus. To the best of our knowledge, we are not aware of other concurrent work on this topic.
> >
> > Reference: Changsheng Wang, Chongyu Fan, Yihua Zhang, Jinghan Jia, Dennis Wei, Parikshit Ram, Nathalie Baracaldo, and Sijia Liu. Reasoning model unlearning: Forgetting traces, not just answers, while preserving reasoning skills. arXiv preprint arXiv:2506.12963, 2025

---

> ### Comment · Reviewer_n7cM · 2025-09-20
> **Thanks for the rebuttal**
>
> Hi authors,
> I think there is still enough time for discussion.
>
> Please use the following code to give a comprehensive evaluation on the four losses.
>
> https://github.com/domenicrosati/representation-noising
>
> https://github.com/rishub-tamirisa/tamper-resistance
>
> https://github.com/git-disl/Booster
>
> Note, all these repos are given in the papers. I don't understand why the authors can't find these repos.
>
> Yes, unlearning loss can be unbounded which make it very instable. In TAR loss they have some optimization on the negative nll loss to make it bounded as well. Please pay attention to this detail. For RepNoise and Booster, the unlearning loss is bounded so they don't have these unbounded issue.  Please also upload the code after your implement and evaluate these three  losses.
>
> Also, I don't think the results you give in the rebuttal is displayed nicely. Please use a table to show the data and discuss it afterwards.

---

> > ### Author Response · Authors · 2025-09-20
> >
> > Thank you for your quick response and for sharing the repository links. We will carefully study these implementations. Within the limited rebuttal window, we will do our best to evaluate the three additional losses and present all the results in a table for clarity.

---

> > > ### Author Response · Authors · 2025-10-06
> > > **Results on other unlearning methods**
> > >
> > > Thank you again for your follow-up comments. We have expanded our comparisons to include additional unlearning methods within our fine-tuning framework. Specifically, we implemented both the **Ascent** loss and the **RepNoise** loss, and their corresponding implementations have been uploaded. For Ascent loss, we use a similar coefficient (1e-5) as unlikelihood loss. For RepNoise, we followed the official implementation, which applies the logarithmic form of the Ascent loss and includes representational regularization. We also adopted their default configuration, where the learning and unlearning losses are assigned equal weights.
> > >
> > > We also made extensive efforts to implement the remaining two unlearning methods, TAR and Booster. For **TAR**, it is originally designed based on the DPO loss, which requires access to reference models. Since our fine-tuning framework is reference-free by design, this approach is not directly compatible with our setting; therefore, we consider it inappropriate to include TAR in our comparison.
> > >
> > > For **Booster**, it is built upon model-agnostic meta-learning (MAML) and involves temporary parameter updates based on gradients. However, we found the original implementation difficult to integrate into our distributed training setup using DeepSpeed ZeRO-3. In particular, the operation `model.named_parameters()` returns zero-sized tensors under ZeRO-3 sharding ([see Booster trainer implementation](https://github.com/git-disl/Booster/blob/main/trainer.py#L297)). We spent several days attempting to resolve this issue but did not find an effective solution to incorporate Booster into the ZeRO-3 framework.
> > >
> > > Nevertheless, we regard TAR and Booster as promising directions for future work and have cited them appropriately in our paper. We have updated the PDF with the latest version reflecting these changes.
> > >
> > >
> > > ## CoT-reasoning results on other unlearning losses
> > >
> > > We evaluated **Ascent** and **RepNoise** on the full Game-of-24 benchmark using Qwen2.5-Math 1.5B and 7B base models, with the same learning rates as in SFT and UFT.
> > > As shown in the results below, **Ascent** achieves performance comparable to SFT and UFT, with slightly lower averages than UFT, consistent with our earlier preliminary findings. **RepNoise**, which places strong emphasis on the unlearning objective (via equal weighting and representation-level forgetting), performs consistently worse than the other methods. These observations suggest that directly applying unlearning methods originally designed for LLM safety domains to reasoning tasks may require additional adaptation and design considerations.
> > >
> > > Moreover, across all unlearning methods, our results consistently support the claim that higher training data quality, regardless of whether the data originate from CoT or search-based algorithms, leads to better fine-tuning performance.
> > >
> > >
> > > **Base LLM: Qwen2.5-Math 7B**
> > >
> > > | Data type (quality) | Metric | SFT | UFT | Ascent | RepNoise |
> > > |----------------------|---------|-----|-----|-----|-----|
> > > | CoT data (92.9%) | succ (greedy) | **28.2%** ± 1.7% | **28.2%** ± 3.6% | 26.8% ± 2.1% | 20.5%  ± 3.7%|
> > > |  | succ (pass@1) | 22.5% ± 3.4% | 23.2% ± 3.4% | **25.5%** ± 2.4% | 17.2%  ± 1.0% |
> > > | ToT+RAP data (69.3%) | succ (greedy) | 13.2% ± 3.6% | **15.2%** ± 2.2% | 13.5% ± 5.1% | 12.8%  ± 3.0% |
> > > |  | succ (pass@1) | 13.2% ± 2.8% | **17.2%** ± 1.0% | 15.8% ± 2.1% | 13.2%  ± 5.1% |
> > > | CoT+ToT+RAP data (95.3%) | succ (greedy) | 27.5% ± 4.2% | **30.2%** ± 2.1% | 28.5% ± 1.7% | 20.8%  ± 2.1% |
> > > |  | succ (pass@1) | 24.2% ± 5.2% | **26.2%** ± 2.2% | 25.5% ± 3.1% | 19.0%  ± 2.2% |
> > >
> > > **Base LLM: Qwen2.5-Math 1.5B**
> > >
> > > | Data type (quality) | Metric | SFT | UFT | Ascent | RepNoise |
> > > |----------------------|---------|-----|-----|-----|-----|
> > > | CoT data (82.1%) | succ (greedy) | 14.5% ± 3.8% | **18.2%** ± 3.2% | 15.8% ± 2.8% | 12.2% ± 3.3% |
> > > |  | succ (pass@1) | 17.8% ± 3.4% | **18.5%** ± 3.5% | 17.0% ± 2.2% | 12.0% ± 2.2% |
> > > | ToT+RAP data (44.7%) | succ (greedy) | **9.5%** ± 0.6% | 8.0% ± 1.4% |  9.2% ± 3.8% | 5.5% ± 0.6% |
> > > |  | succ (pass@1) | **10.0%** ± 1.4% | 9.5% ± 2.6% |9.0% ± 1.8% | 6.2% ± 1.5% |
> > > | CoT+ToT+RAP data (87.6%) | succ (greedy) | 20.0% ± 2.7% | 20.5% ± 3.7% | **20.8%** ± 2.1% | 10.3% ± 3.6% |
> > > |  | succ (pass@1) | 17.0% ± 2.2% | 18.8% ± 1.3% | **20.0%** ± 3.4% | 12.8% ± 3.6% |

---

> > > > ### Author Response · Authors · 2025-10-06
> > > > **Continued**
> > > >
> > > > ## Search-based reasoning results on other unlearning losses
> > > >
> > > > Lastly, following the setup described in our paper, we evaluate LLMs fine-tuned with CoT traces on their ability to maintain search-based reasoning performance. For the 7B base model, we sweep the learning rate over {1e-6, 2e-6, 5e-6}.
> > > >
> > > > We observe that for **RepNoise**, regardless of the learning rate, the search-based performance (ToT and RAP) drops to 0.0, even though its CoT-reasoning performance improves. This suggests that applying strong unlearning on failed CoT traces can overly suppress useful intermediate reasoning patterns, thereby harming the model’s ability to perform effective search.
> > > >
> > > > For **Ascent**, our main findings for SFT and UFT remain valid, as shown in the results below. We highlight the best results for each pair of training data and inference method.
> > > >
> > > > * Learning rate effect: a smaller learning rate slightly reduces CoT reasoning performance but can substantially preserve, or even improve, the base model’s search capability (e.g., 31% in ToT and 28% in RAP).
> > > > * CoT training is more effective for maintaining search ability compared to mixed data (CoT+ToT+RAP), suggesting that using more on-policy data reduces distribution shift and mitigates forgetting, as discussed in the paper.
> > > >
> > > >
> > > > | Data type (quality)  | Learning rate | CoT (greedy) | ToT | RAP |
> > > > |----------------------|---------|-----|-----|-----|
> > > > | Base LLM | N/A | 6.0% | 28.0% | 27.0% |
> > > > | CoT data (92.9%) | 5e-6 |  **26.8%** ± 2.1% | 15.5% ± 4.1% | 11.2% ± 1.7% |
> > > > | | 2e-6 | 23.0% ± 0.8% | **31.0%** ± 8.8% | 22.8% ± 4.5% |
> > > > | | 1e-6 | 20.2% ± 1.0% | 24.2% ± 3.5% | **28.0%** ± 2.6% |
> > > > | ToT+RAP data (69.3%) | 5e-6 | **13.5%** ± 5.1% | 6.8% ± 2.9% | 6.2% ± 2.2% |
> > > > | | 2e-6 | 9.8% ±  1.5% |  20.0% ± 0.8% | **18.8%** ± 7.6% |
> > > > | | 1e-6 | 11.8% ±  1.7% | **20.8%** ± 5.9% | 18.5% ± 3.3% |
> > > > | CoT+ToT+RAP data (95.3%) | 5e-6 | **28.5%** ± 1.7% | 14.8% ± 2.5% | 12.0% ± 1.8% |
> > > > | | 2e-6 | 20.2% ± 1.9% | **29.5%** ± 3.3% | **21.8%** ± 2.5% |
> > > > | | 1e-6 | 17.5% ± 1.3% | 23.8% ± 3.8% | 13.5% ± 3.1% |
> > > >
> > > > We hope these additional results address your concerns. Please let us know if there are any remaining questions or points that need clarification.

---

> ### Comment · Reviewer_n7cM · 2025-10-10
> **This result is rather interesting**
>
> Hi authos,
>
> Thanks for the additional results. The rebuttal results surprises me a lot:
>
>
> Particularly, it is surprising to see that RepNoise completely destroy the search-based performance (ToT and RAP) but the performance after COT trainining still preserve (if my understand correctly). Could you please give some insights on that: why the model still preserve reasoning performance but after applying  search-based strategic it completely destroy?   In my understanding, ToT and RAP just apply multiple inference on the model and if the reasoning ability of the model still preserve, it should not make the accuracy to strictly 0.
>
> Also, please update the result of RepNoise on COT, ToT and RAP into a table and discuss your conjecture on why it fails? The conjecture does not have to be correct but I think this phenomenon is generally very interesting,
>
>
> Feel free to say your real conjecture because I will give strong acceptance of this paper anyway, as I do like the extra finding and the paper itself when I read it.  For Booster and TAR, I am super curious about their performance in this task as well.
> For Booster, could you disable deepspeed zero3 for it (use original ddp) and try to compare?

---

> ### Author Response · Authors · 2025-10-11
> **Search-based reasoning results on RepNoise and more explanation on unlearning losses**
>
> Thank you very much for your positive feedback and for highlighting this intriguing phenomenon.
>
> Following your suggestion, we report the full search-based results on **RepNoise** below. Your understanding is correct: for RepNoise, regardless of the learning rate, the search-based performance (ToT and RAP) drops to 0.0%, even though CoT reasoning performance improves over the base LLM.
>
> | Data type (quality)  | Learning rate | CoT (greedy) | ToT | RAP |
> |----------------------|---------|-----|-----|-----|
> | Base LLM | N/A | 6.0% | 28.0% | 27.0% |
> | CoT data (92.9%) | 5e-6 |  **20.5%** ± 3.7% | 0.0% ± 0.0% | 0.0% ± 0.0% |
> | | 2e-6 | 13.5% ± 0.6% | 0.0% ± 0.0% | 0.0% ± 0.0% |
> | | 1e-6 | 8.0% ± 0.8% | 0.0% ± 0.0% | 0.0% ± 0.0% |
> | ToT+RAP data (69.3%) | 5e-6 | **12.8%** ± 3.0% | 0.0% ± 0.0%| 0.0% ± 0.0% |
> | | 2e-6 | 8.2% ±  1.3% |  0.0% ± 0.0% | 0.0% ± 0.0% |
> | | 1e-6 | 6.0% ±  1.4% | 0.0% ± 0.0% | 0.0% ± 0.0% |
> | CoT+ToT+RAP data (95.3%) | 5e-6 | **20.8%** ± 2.1% | 0.0% ± 0.0% | 0.0% ± 0.0% |
> | | 2e-6 | 17.8% ± 1.0% | 0.0% ± 0.0% | 0.0% ± 0.0% |
> | | 1e-6 | 14.5% ± 3.4% | 0.0% ± 0.0% | 0.0% ± 0.0% |
>
> **Our conjecture.** We offer two main hypotheses for why RepNoise leads to complete search degradation despite improved CoT reasoning:
> - The default coefficient of 0.5 in the RepNoise implementation substantially increases the *effective* learning rate, much larger than that used in other unlearning methods like UFT and Ascent ($\le$ 1e-3). Even at small learning rates, this strong gradient signal may lead to severe over-updating during unlearning.
> - RepNoise applies representational regularization toward random noise, which may induce stronger parameter changes. We hypothesize that this disrupts internal representations necessary for search (e.g., hypothesis branching, reward estimation), even though the training task (CoT) remains intact.
>
> We believe this highlights an important and underexplored aspect of LLM unlearning for reasoning: designing objectives that remove unwanted behaviors without destroying useful internal structure. In our framework, where models are trained on CoT traces but evaluated under search-based inference, this is particularly critical. With appropriate learning rates and coefficients, unlearning can indeed improve search inference (as shown in Figure 4). However, overly strong unlearning signals likely cause catastrophic forgetting of pretrained knowledge that ToT and RAP rely on.
>
> Regarding the difference between CoT and **ToT/RAP**: these methods fundamentally differ in inference style. CoT generates a single plausible reasoning path end-to-end, whereas ToT/RAP require (1) generating multiple *intermediate* reasoning paths, and (2) evaluating their success likelihood as a reward signal (see Appendix D for details). Thus, it is possible to be strictly zero success rate for ToT/RAP, e.g., all intermediate paths are wrong in the format (the LLM does not follow the prompt but instead outputs entire reasoning paths). This more complex reasoning is likely more sensitive to representational disruption, explaining why ToT/RAP performance collapses while CoT remains functional.
>
> Finally, regarding the implementation of **Booster**, we believe it is possible to run it by disabling Zero-3, although this would significantly increase training time. Our conjecture is that, with the default loss coefficients and learning rates, Booster may experience a similar degradation phenomenon to RepNoise. In other words, achieving competitive results would likely require a careful hyperparameter sweep over both the loss coefficients and learning rates. Conducting such an extensive exploration is beyond the scope of our current work, which focuses on demonstrating that unlearning can benefit reasoning, an underexplored direction in prior work. We view unlikelihood training as one effective approach to this, while more advanced unlearning methods with thorough hyperparameter tuning remain promising directions for future research.
>
> We thank the reviewer again for the valuable feedback and encouragement.

---

### Author Response · Authors · 2025-09-09
**Planned Updates**

We thank both reviewers for their thoughtful feedback. In response, we are conducting additional mathematical experiments as suggested. We will update the paper with the new results and provide a more complete revision in the coming weeks.

---

### Decision · Action_Editor_jrTu · 2025-10-10

**Recommendation:** Accept as is

**Audience:**

Yes

**Audience Explanation:**

Yes. The findings would be of clear interest to the TMLR audience, particularly researchers working on reasoning, fine-tuning, and efficiency in large language models.

**Claims And Evidence:**

Yes

**Claims Explanation:**

Yes. The claims are supported by clear and convincing experimental evidence. The proposed method is well justified and validated through consistent improvements on challenging reasoning benchmarks. The results are coherent with the stated objectives and demonstrate the effectiveness of the approach.